# Representation Unlearning: Forgetting through Information Compression

Antonio Almudévar [1]    Alfonso Ortega [1]

## Abstract

Machine unlearning seeks to remove the influence of specific training data from a model, a need driven by privacy regulations and robustness concerns. Existing approaches typically modify model parameters, but such updates can be unstable, computationally costly, and limited by local approximations. We introduce Representation Unlearning, a framework that performs unlearning directly in the model's representation space. Instead of modifying model parameters, we learn a transformation over representations that imposes an information bottleneck: maximizing mutual information with retained data while suppressing information about data to be forgotten. We derive variational surrogates that make this objective tractable and show how they can be instantiated in two practical regimes: when both retain and forget data are available, and in a zero-shot setting where only forget data can be accessed. Experiments across several benchmarks demonstrate that Representation Unlearning achieves more reliable forgetting, better utility retention, and greater computational efficiency than parameter-centric baselines.

## 1. Introduction

Modern machine learning systems are increasingly trained on massive, continuously evolving datasets that contain sensitive, personal, or potentially problematic information. As a consequence, there is a growing need for models that can forget specific training examples or subsets of data upon request. This requirement is amplified by regulatory frameworks such as the European Union GDPR (Mantelero, 2013) and California Consumer Privacy Act (CCPA) (Pardau, 2018), which enshrine a "right to be forgotten" (Dang, 2021); and by practical considerations such as the removal of mislabeled samples (Northcutt et al., 2021), poisoned

data (Steinhardt et al., 2017), or users who withdraw consent (Politou et al., 2018a). Simply deleting the raw data is insufficient: deep networks are known to memorize training information (Arpit et al., 2017; Carlini et al., 2019; Feldman, 2020), and their parameters often retain statistical traces of the removed samples (Melis et al., 2019; Bourtoule et al., 2021). This has motivated a surge of interest in machine unlearning, the problem of efficiently removing the influence of designated data from a trained model (Ginart et al., 2019). More concretely, effective unlearning requires balancing two key criteria: (i) verifiability—ensuring that forget data cannot be reconstructed or detected, and (ii) utility—preserving performance on the retained data.

Naively retraining a model from scratch after each deletion request guarantees perfect unlearning but is typically computationally prohibitive, especially in large-scale settings. This has motivated a growing body of work on efficient alternatives that approximate the effect of retraining without its full cost. Most existing approaches adopt a parameter-centric perspective, directly modifying model weights to remove the influence of deleted samples (Bourtoule et al., 2021; Neel et al., 2021). While effective in some settings, this approach faces several limitations. First, many methods exhibit poor performance scalability, struggling to preserve model utility when applied to large-scale architectures and complex datasets (Basu et al., 2020; Xu et al., 2023). Second, parameter updates often require costly second-order information or multiple backward passes, narrowing their efficiency advantage over naive retraining (Sekhari et al., 2021). Third, the requirement to potentially modify the entire parameter space results in heavy GPU usage, creating a computational bottleneck that scales poorly as model sizes increase (Gao et al., 2024; Mittal, 2025).

In this work, we introduce *Representation Unlearning*, a framework that departs from parameter-centric methods by operating directly in the model's representation space. Rather than modifying or re-optimizing network parameters, our approach targets the internal representations, where the influence of individual samples is more localized and easier to control (Bengio et al., 2013; Bau et al., 2017; Mu & Andreas, 2020). *Representation Unlearning* learns a transformation $f_\phi$ that maps the original representation $Z$ to a new one $Z'$, enforcing an information bottleneck (Tishby et al., 2000): the transformed representation is encouraged

---

[1]ViVoLab, Aragón Institute for Engineering Research (I3A), University of Zaragoza, Zaragoza, Spain. Correspondence to: Antonio Almudévar <almudevar@unizar.es>.

*Proceedings of the 43rd International Conference on Machine Learning*, Seoul, South Korea. PMLR 306, 2026. Copyright 2026 by the author(s).

to retain high mutual information with the data in the retain set $\mathcal{D}_r$ while suppressing mutual information with the data in the forget set $\mathcal{D}_f$, as illustrated in Figure 1. This information-theoretic objective enables targeted removal of undesired influences in the representation and preserving performance on the remaining data while sidestepping the instability and computational cost associated with parameter-level unlearning.

As these mutual-information terms are generally intractable, we introduce variational approximations that yield efficient, optimizable surrogates (Alemi et al., 2016; Poole et al., 2019). We then show how these surrogates can be instantiated under two complementary data-access regimes: (i) a setting where both retain and forget data are available; and (ii) a zero-shot scenario in which only the forget data can be accessed (Chundawat et al., 2023b). The first setting is the one implicitly assumed by most existing unlearning methods, as access to both subsets simplifies optimization (Ginart et al., 2019; Bourtoule et al., 2021). However, the second scenario, while more challenging, is equally important in practice: in many real-world applications, access to retained data may be restricted due to privacy, storage limitations, or organizational constraints (Politou et al., 2018b; Liu et al., 2020). Addressing both regimes is therefore essential for developing unlearning methods that remain applicable beyond idealized experimental conditions.

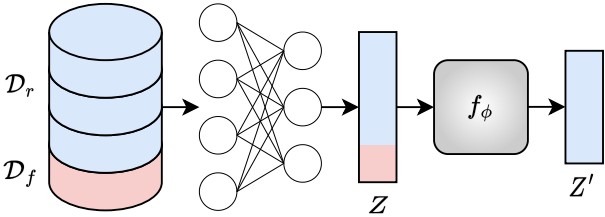

*Figure 1.* The model is first trained on both the retain set $\mathcal{D}_r$ (blue) and the forget set $\mathcal{D}_f$ (red), leading the learned representation $Z$ to encode information from both. *Representation Unlearning* seeks to learn a transformation $f_\phi$ that maps $Z$ to a new representation $Z'$ in which information attributable to $\mathcal{D}_f$ is removed while information relevant to $\mathcal{D}_r$ is preserved.

## 2. Related Work

**Baselines and Exact Methods in Machine Unlearning** The gold standard, **Retraining** ($S_{\text{retrain}}$), guarantees exact unlearning by training from scratch on the retain set but is computationally prohibitive. The **Baseline** represents the original model trained on all data. Its naive approximation, **Fine-tuning** on the retain set, is efficient but often fails to fully erase information or suffers from catastrophic forgetting (Kirkpatrick et al., 2017). **SISA** (Bourtoule et al., 2021) achieves exact unlearning by partitioning data into shards and retraining only the affected sub-models, though this incurs high storage and inference latency.

**Unlearning with Retain Data Access** Most methods leverage retain data to preserve utility, typically by updating model parameters. These approaches generally fall into five categories based on their core mechanisms. Distillation frameworks, such as **SCRUB** (Kurmanji et al., 2023) and **Bad Teacher** (Chundawat et al., 2023a), use teacher networks to balance forget-set error with retain-set knowledge. Gradient manipulation and noise injection methods reverse the learning trajectory; for example, **Amnesiac** (Graves et al., 2021) and **UNSIR** (Tarun et al., 2023) subtract gradients or maximize noise, while **Langevin Unlearning** (Chien et al., 2024) applies noisy gradient descent. Adversarial methods like **AMUN** (Ebrahimpour-Boroojeny et al., 2025) overwrite target concepts using adversarial examples. Conversely, information injection approaches like **NatMU** (He et al., 2025) induce forgetting by blending retain features directly into the forget set. Finally, parameter masking and information-theoretic methods—including **SSD** (Foster et al., 2024), **Fisher Forgetting**, and **Variational Unlearning** (Golatkar et al., 2020a;b)—protect crucial retain weights using metrics like the Fisher Information Matrix. Despite their diverse mechanics, all these techniques are fundamentally parameter-centric. Consequently, their performance degrades significantly as complexity scales. By operating strictly in the representation space, our framework bypasses unstable weight updates to maintain consistent efficacy across complex tasks.

**Zero-Shot and Forget-Only Unlearning** In the restricted setting without retain data, **Unroll** (Thudi et al., 2022) approximates the inverse training trajectory. Methods like **GKT** and **EMMN** (Chundawat et al., 2023b) rely on generative models or prototypes to hallucinate retain samples, while **Boundary Shrinking** (Chen et al., 2023) directly shifts the decision boundary of the forget class.

**Concept Erasure and Representation Editing** Our framework relates to concept erasure methods, such as LEACE (Belrose et al., 2023) and kernelized rate-distortion (Basu Roy Chowdhury et al., 2023), which project out specific attributes (e.g., protected traits). It also parallels feature-space unlearning techniques like domain adversarial training (Sepahvand et al., 2025) and gradient-free forgetting (Kodge et al., 2024). However, our approach differs fundamentally. First, instead of erasing a shared attribute using explicit concept labels, we remove the holistic identity and influence of entire instances or classes. Second, rather than relying on complex adversarial objectives, heavy kernel estimators, or iterative projections, we train a lightweight transformation $f_\phi$ guided by tractable information-theoretic bounds. Finally, by exploiting Neural Collapse, our formulation uniquely enables zero-shot unlearning without retain data—a capability absent in standard concept erasure.

# 3. Representation Unlearning

## 3.1. Problem Description

We consider a learning setting in which a model receives inputs $X \in \mathcal{X}$ and is trained to predict targets $Y \in \mathcal{Y}$. The model is trained on a dataset $\mathcal{D} = \{(x^{(i)}, y^{(i)})\}_{i=1}^{N}$, which induces the empirical distribution $p(x,y) \approx \frac{1}{N} \sum_{i=1}^{N} \delta(x - x^{(i)})\,\delta(y - y^{(i)})$. Given an input $x$, the model produces a prediction $\hat{y}$ according to the conditional distribution $p_\theta(\hat{y} \mid x)$, where $\theta \in \Theta$ denotes the *original* model parameters. In addition, the model computes intermediate representations $z \in \mathcal{Z}$, which we denote by the distribution $p_\theta(z \mid x) = \delta(z - e_\theta(x))$, where $e_\theta$ is the encoder.

The goal of machine unlearning is to obtain a model that behaves as if a designated subset of the training data, the forget set $\mathcal{D}_f \subset \mathcal{D}$, had never been used during training—or equivalently, as if the model had been trained solely on the retain set $\mathcal{D}_r = \mathcal{D} \setminus \mathcal{D}_f$. In this case, we consider the inputs $X_r \in \mathcal{X}$ and targets $Y_r \in \mathcal{Y}$ corresponding to the retained data, with $\mathcal{D}_r = \{(x_r^{(i)}, y_r^{(i)})\}_{i=1}^{N_r}$ inducing the empirical distribution $p(x_r, y_r) \approx \frac{1}{N_r} \sum_{i=1}^{N_r} \delta(x_r - x_r^{(i)})\,\delta(y_r - y_r^{(i)})$. The model we would obtain produces predictions according to $p_{\theta_r}(\hat{y}_r \mid x_r)$, where $\theta_r \in \Theta$ denotes the parameters of this *forgetting* model. It also computes intermediate representations $z_r \in \mathcal{Z}$ via the distribution $p_{\theta_r}(z_r \mid x_r)$.

## 3.2. Introducing the Transformation

Most existing approaches aim to construct endomorphisms $g_\psi : \Theta \to \Theta$ that modify the model parameters so as to obtain $\theta' = g_\psi(\theta) \approx \theta_r$. However, because modern neural networks contain millions (or even billions) of parameters, the parameter space $\Theta$ is extremely high-dimensional. Learning such mappings is therefore challenging, expensive, and often unstable. In contrast, *representation unlearning* operates directly in the representation space. Instead of modifying parameters, it learns a mapping $f_\phi : \mathcal{Z} \to \mathcal{Z}$ such that $z' = f_\phi(z) \approx z_r$. Since the representation space $\mathcal{Z} \subseteq \mathbb{R}^d$ typically has dimension that is orders of magnitude smaller than the number of model parameters, the transformation $f_\phi$ is substantially simpler and more tractable to learn than the parameter-level mapping $g_\psi$. We next characterize the conditions that the transformation $f_\phi$ must satisfy for $Z'$ to remain informative about $\mathcal{D}_r$ while eliminating information pertaining to $\mathcal{D}_f$.

**On the architecture of** $f_\phi$    We designate $Z$ as the activations of the penultimate layer. Leveraging the "Linear Representation Hypothesis"—which posits that semantic information becomes *almost linearly separable* at deeper layers (Alain & Bengio, 2016; Raghu et al., 2017; Conneau et al., 2018; Nanda et al., 2023; Park et al., 2023)—we constrain $f_\phi$ as a lightweight linear map or shallow MLP. This

approach ensures computational efficiency without sacrificing expressivity, as we demonstrate in Section 4.

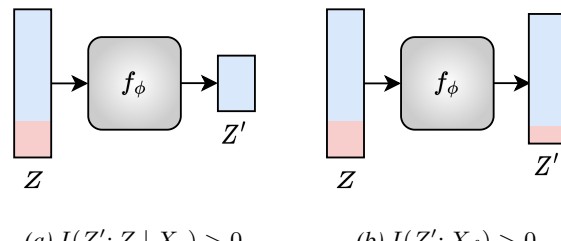

*(a) $I(Z'; Z \mid X_r) > 0$          (b) $I(Z'; X_f) > 0$*

*Figure 2.* $I(Z'; Z \mid X_r) > 0$ indicates that the transformation $f_\phi$ removes some information relevant to $X_r$. Conversely, $I(Z'; X_f) > 0$ indicates that $f_\phi$ does not fully eliminate information associated with $X_f$.

## 3.3. How to Retain Information about $\mathcal{D}_r$

The first objective of machine unlearning is to preserve performance on the retain set. In our information-theoretic formulation, this requires that the transformed representation $Z'$ retains all information about $X_r$ that is present in the original representation $Z$. In other words, applying the transformation $f_\phi$ on $Z$ should not discard any information relevant to the retained data. Formally, this requirement can be expressed as

$$I(Z'; Z \mid X_r) = 0. \tag{1}$$

Figure 2a illustrates the deviation of $Z'$ when this condition is violated. Exploiting the non-negativity of mutual information, we frame this as a minimization problem. However, direct optimization is intractable due to the integration over high-dimensional input and representation spaces. To circumvent this, we derive the following variational upper bound in Appendix A.1:

$$\mathcal{L}_r \coloneqq \mathbb{E}_{x_r \sim p(x_r)}\Big[\mathbb{E}_{z \sim p_\theta(z|x_r)}\big[$$
$$D_{\mathrm{KL}}(p_\phi(z' \mid z)\,\|\,r_\theta(z' \mid x_r))\big]\Big] \geq I(Z'; Z \mid X_r), \tag{2}$$

where $p_\phi(z' \mid z)$ denotes the distribution induced by the transformation and $r_\theta(z' \mid x_r)$ is a variational approximation of $p_{\theta,\phi}(z' \mid x_r)$. Computing Equation 2 presents two challenges: (i) the lack of closed-form KL expressions for arbitrary distributions and (ii) the intractability of high-dimensional integration. We address these via parameterization and sampling, respectively.

First, since our goal is to leave the retain samples unaltered, we define $r_\theta(z' \mid x_r)$ to be as close as possible to the original encoder distribution $p_\theta(z \mid x_r)$. However, treating the encoder as a deterministic mapping—i.e., as a Dirac delta—leads to undefined KL divergence terms. To address this, we adopt a probabilistic formulation by modeling $r_\theta$ as a Gaussian distribution centered at the encoder output $e_\theta(x)$.

Similarly, we model $p_\phi$ as a Gaussian as well, following standard practice (Bishop, 2006).

$$r_\theta(z' \mid x) = \mathcal{N}(e_\theta(x), I_d), \tag{3}$$

$$p_\phi(z' \mid z) = \mathcal{N}(f_\phi(z), I_d). \tag{4}$$

where $I_d$ is the $d \times d$ identity matrix.

Second, we approximate the intractable integrals via Monte Carlo estimation. Leveraging the Gaussian parameterization—under which the KL divergence reduces to the squared Euclidean distance (i.e., mean squared error)—we arrive at the following tractable objective:

$$\mathcal{L}_r \approx \frac{1}{B_r M} \sum_{i=1}^{B_r} D_{\text{KL}}\big(p_\phi(z' \mid z^{(i)}) \,\|\, r_\theta(z' \mid x_r^{(i)})\big)$$

$$= \frac{1}{2B_r} \sum_{i=1}^{B_r} \big\| z_r^{(i)} - f_\phi(z_r^{(i)}) \big\|_2^2. \tag{5}$$

where $z_r^{(i)} = e_\theta(x_r^{(i)})$ and $x_r^{(i)} \sim p(x_r)$. Intuitively, this loss acts as a consistency constraint, penalizing any deviation of the transformed representations from their original topological structure for the retained data.

**Zero-shot setting** Evaluating Eq. 2 requires sampling from the data marginal $p(x_r)$, which is prohibited in the zero-shot setting. To overcome this, we derive a relaxed variational upper bound that relies on label marginalization rather than input sampling. We assume access to basic training metadata—specifically the per-class counts $N^c$—allowing us to estimate the prior class probability as $p(y = c) \approx N^c/N$.

In Appendix A.2, we show that by relaxing the conditioning from inputs $X_r$ to labels $Y_r$, we obtain the following tractable upper bound for $I(Z'; Z \mid X_r)$:

$$\mathcal{L}_r^{zs} := \mathbb{E}_{y_r \sim p(y_r)} \Big[ \mathbb{E}_{z \sim p_\theta(z \mid y_r)} \big[ \tag{6}$$

$$D_{\text{KL}}(p_\phi(z' \mid z) \,\|\, r_\theta(z' \mid y_r)) \big] \Big] \geq I(Z'; Z \mid X_r).$$

where $r_\theta(z' \mid y_r)$ is a variational approximation of $p_{\theta,\phi}(z' \mid y_r)$. Operationalizing this bound presents three challenges: (i) estimating the class prior $p(y_r)$, (ii) approximating the class-conditional density $p_\theta(z|y_r)$ without data, and (iii) finding an accurate variational distribution $r_\theta(z' \mid y_r)$.

To resolve the first challenge, we invoke the *Law of Total Probability*, decomposing the global label marginal $p(y)$ into a mixture of retain and forget distributions weighted by their respective sample counts. By combining the total training size $N$ with the known forget set size $N_f$, we isolate and recover the retain prior $p(y_r)$ as follows:

$$p(y_r = c) = \frac{N p(y = c) - N_f p(y_f = c)}{N - N_f} = \frac{N_r^c}{N_r}. \tag{7}$$

For the second and third challenges, we leverage the *Neural Collapse* hypothesis (Papyan et al., 2020). This posits that, in the terminal phase of training, within-class feature variability collapses and class means converge to their corresponding linear classifier weights. Thus, we approximate $p_\theta(z \mid y = c)$ by the weight vector $w_c$, and model $r_\theta(z' \mid y = c)$ as an isotropic Gaussian centered at $w_c$:

$$p_\theta(z \mid y = c) \approx \delta(z - w_c), \tag{8}$$

$$r_\theta(z' \mid y = c) \approx \mathcal{N}(z'; w_c, I_d). \tag{9}$$

This enables the sampling of "pseudo-targets" that inherently respect the original class imbalance, relying solely on model parameters and metadata.

Finally, mirroring the data-driven approximation, we estimate $\mathcal{L}_r^{zs}$ via Monte Carlo sampling. We iterate over the $C$ classes, drawing synthetic latent samples from the Neural Collapse proxies weighted by the derived retain prior. This yields the following empirical zero-shot retention loss:

$$\mathcal{L}_r^{zs} \approx \frac{1}{2N_r} \sum_{c=1}^{C} N_r^c \big\| w_c - f_\phi(w_c) \big\|_2^2, \tag{10}$$

where $N_r^c$ denotes the cardinality of class $c$ within the retain set. Intuitively, this serves as a semantic anchor, preserving the latent structure by penalizing deviations from the class centroids $w_c$.

### 3.4. How to Forget Information about $\mathcal{D}_f$

Complementing the retention objective, effective unlearning demands the erasure of information regarding the forget set $X_f$ from the transformed representation $Z'$. Formally, this requires that $Z'$ be statistically independent of $X_f$:

$$I(Z'; X_f) = 0. \tag{11}$$

Figure 2b visualizes the failure to meet this condition. Since direct minimization of mutual information is intractable, we use a variational upper bound (derived in Appendix A.3):

$$I(Z'; X_f) \leq \mathbb{E}_{p(x_f)} \left[ D_{\text{KL}} \left( p_{\theta,\phi}(z' \mid x_f) \,\|\, r_\theta(z') \right) \right] \tag{12}$$

Evaluating this expression faces the same obstacles regarding the lack of closed-form KL solutions. We resolve this by expanding the definitions of the aggregate transition and marginal distributions:

$$p_{\theta,\phi}(z' \mid x_f) = \int p_\phi(z'|z) p_\theta(z|x_f) \, dz, \tag{13}$$

$$r_\theta(z') = \int r_\theta(z'|x) p(x) \, dx. \tag{14}$$

Relying on the joint convexity of the KL divergence, we apply Jensen's inequality (Jensen, 1906) to upper-bound the

divergence term (derivation provided in Appendix A.4):

$$D_{\text{KL}}\Big(p_{\theta,\phi}(z' \mid x_f) \,\Big\|\, r_\theta(z')\Big) \leq \mathbb{E}_{z \sim p_\theta(z|x_f)}\Big[$$
$$\mathbb{E}_{x \sim p(x)}\big[D_{\text{KL}}\big(p_\phi(z'|z) \,\|\, r_\theta(z'|x)\big)\big]\Big]. \qquad (15)$$

Substituting this result back into Equation 12, we obtain the final variational upper bound $\mathcal{L}_f$:

$$\mathcal{L}_f := \mathbb{E}_{x_f \sim p(x_f)}\Big[\mathbb{E}_{z \sim p_\theta(z|x_f)}\Big[\mathbb{E}_{x \sim p(x)}\big[$$
$$D_{\text{KL}}\big(p_\phi(z'|z) \,\|\, r_\theta(z'|x)\big)\big]\Big]\Big] \geq I(Z'; X_f). \qquad (16)$$

Proceeding analogously to the retention objective, we estimate $\mathcal{L}_f$ via Monte Carlo sampling. Leveraging the result from Eq. 5—where the Gaussian parameterization reduces the KL divergence to the mean squared error—we sample a mini-batch of $B_f$ forget instances and compare them against a reference batch of size $B$ drawn from the marginal $p(x)$ (spanning both retain and forget data). This yields the following empirical loss:

$$\mathcal{L}_f \approx \frac{1}{2B_f B} \sum_{i=1}^{B_f} \sum_{j=1}^{B} \big\| z^{(j)} - f_\phi(z_f^{(i)}) \big\|_2^2, \qquad (17)$$

where $z^{(j)} = e_\theta(x^{(j)})$, $x^{(j)} \sim p(x)$, $z_f^{(i)} \sim e_\theta(x_f^{(i)})$ and $x_f^{(i)} \sim p(x_f)$. Intuitively, this objective forces the transformed representations of the forget set to match the aggregate statistics of the entire dataset, effectively rendering the unlearned representations indistinguishable from the general population.

**Zero-shot setting** Evaluating Eq. 16 similarly requires sampling from the data marginal $p(x)$, which necessitates access to the retain set. To adapt our framework to the zero-shot setting, we approximate the variational approximation $r_\theta(z')$—appearing in Eq. 12—by marginalizing over the label space $\mathcal{Y}$ instead of the input space $\mathcal{X}$:

$$r_\theta(z') = \sum_{y \in \mathcal{Y}} r_\theta(z'|y)p(y). \qquad (18)$$

Mirroring the derivation of the data-driven objective, we invoke Jensen's inequality on the KL functional to derive the zero-shot variational upper bound in Appendix A.5:

$$\mathcal{L}_f^{zs} := \mathbb{E}_{x_f \sim p(x_f)}\Big[\mathbb{E}_{z \sim p_\theta(z|x_f)}\Big[\mathbb{E}_{y \sim p(y)}\big[$$
$$D_{\text{KL}}\big(p_\phi(z'|z) \,\|\, r_\theta(z'|y)\big)\big]\Big]\Big] \geq I(Z'; X_f). \quad (19)$$

To render this tractable, we reuse the proxies established for the retain objective: (i) we set the prior $p(y)$ to the empirical

class frequency $N^c/N$, and (ii) model the class-conditional density $r_\theta(z'|y)$ using the Neural Collapse-based Gaussian approximation (Eq. 9). This allows us to sample "pseudo-targets" from the global distribution without forward passes. By substituting these proxies into the bound, we derive the empirical zero-shot forget loss:

$$\mathcal{L}_f^{zs} \approx \frac{1}{2B_f N} \sum_{i=1}^{B_f} \sum_{c=1}^{C} N^c \big\| w_c - f_\phi(z_f^{(i)}) \big\|_2^2, \qquad (20)$$

where $z_f^{(i)} = e_\theta(x_f^{(i)})$ and $x_f^{(i)} \sim p(x_f)$. Intuitively, by minimizing the weighted distance to all class prototypes $w_c$, this objective forces the transformed forget representations to converge toward the global centroid of the label space. This effectively "drowns" the specific instance information into the aggregate statistics of the original training distribution.

### 3.5. Optimization Objective

To optimize the unlearning transformation $f_\phi$, we minimize a composite objective. In the standard setting with data access, the total loss is defined as:

$$\mathcal{L} = \mathcal{L}_r + \beta \mathcal{L}_f, \qquad (21)$$

where $\beta > 0$ explicitly controls the trade-off between unlearning strength and information retention.

In the **zero-shot setting**, we simply substitute these terms with their corresponding proxies derived above: $\mathcal{L}_r^{zs}$ (Eq. 10) and $\mathcal{L}_f^{zs}$ (Eq. 20). We provide complete pseudo-code for both procedures in Appendix B.

## 4. Experiments

### 4.1. Mechanistic Insight: Unlearning a 2D Space

To visualize the unlearning process without introducing artifacts from dimensionality reduction, we design a controlled experiment using a native 2-dimensional bottleneck. Specifically, we train a 2-layer MLP encoder on synthetic 10-dimensional data spanning six classes, constraining the model to learn a representation space $Z \in \mathbb{R}^2$ in which the classes are linearly separable. One class—depicted as yellow stars—is designated as the target to be forgotten. A detailed description of the dataset is provided in Appendix C.

As shown in Figure 3, the Original Space (Left) reveals the forget class as a distinct, separable cluster. The Standard Unlearning (Center) panel demonstrates that with access to retain data, the forget cluster collapses entirely into the adjacent manifold of neighboring classes (orange/purple). Crucially, the surrounding retain clusters remain rigid, maintaining their original geometry with minimal disturbance. In the Zero-Shot (Right) setting, the method achieves a remarkably similar result: guided solely by Neural Collapse

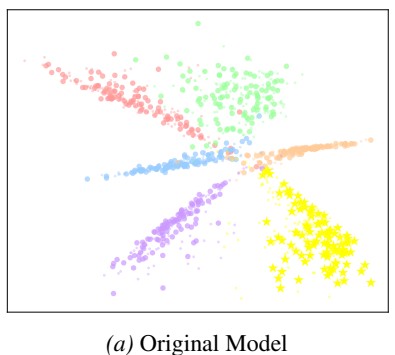 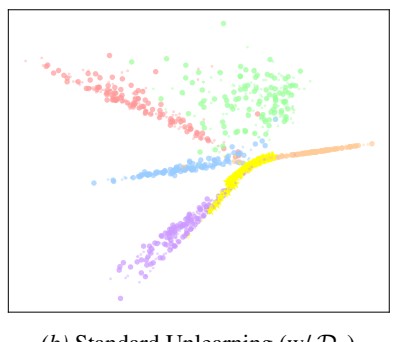 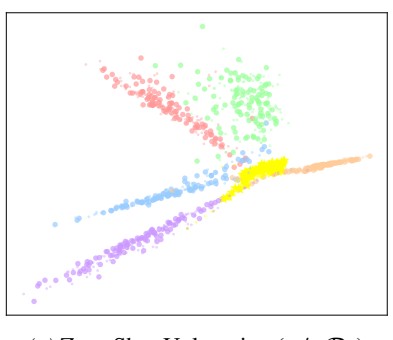

*(a)* Original Model       *(b)* Standard Unlearning (w/ $\mathcal{D}_r$)       *(c)* Zero-Shot Unlearning (w/o $\mathcal{D}_r$)

*Figure 3.* **Mechanistic Comparison of Unlearning Regimes. (a)** The original space shows the forget class (**yellow stars**) as a distinct cluster. **(b)** With retain data, the forget samples collapse into the adjacent manifold while retain clusters remain stable. **(c)** In the zero-shot setting, the forget class exhibits a similar collapse, validating the proxy anchors, albeit with slightly increased modification to the retain representations due to the absence of real data constraints.

proxies, the forget samples are compressed into the same manifold. However, without real retain samples to anchor the space, the remaining clusters exhibit slightly higher distortion, though the global topology remains largely intact.

### 4.2. Experimental Setup

**Datasets and Models**  We evaluate our framework on three standard benchmarks: CIFAR-10, CIFAR-100, and Tiny ImageNet, using ResNet-18, WideResNet-28-10, and ResNet-34 architectures, respectively. We consider two distinct unlearning scenarios: (i) **Class Unlearning**, where we target the removal of 1, 5, and 10 classes for CIFAR-10, CIFAR-100, and Tiny ImageNet, respectively; and (ii) **Random Unlearning**, where the model must forget a randomly selected subset of 10% of the training data.

**Baselines**  We compare Representation Unlearning against three categories of methods: (i) **Reference Baselines**, employing a "do nothing" baseline and Fine-tuning as naive lower bounds, alongside Retraining as the gold standard; (ii) **Standard Setting**, comparing efficiency against SISA, SCRUB, UNSIR, Bad Teacher, Langevin Unlearning, AMUN, and NatMU; and (iii) **Zero-Shot Setting**, comparing our zero-shot approach against Boundary Shrinking, EMMN and GKT.

**Hyperparameters Selection**  For our framework, we set the regularization strength to $\beta = 10^{-3}$ across all experiments. Regarding the transformation architecture, we employ a linear mapping for the **Standard Setting** and a 1-hidden-layer network for the **Zero-Shot Setting**, a configuration validated by the ablation study in Section 4.5. The specific hyperparameters for all competing baselines are detailed in Appendix D.

**Evaluation Protocol**  All results report the mean and standard deviation across 5 random seeds. Class unlearning removes the same class across all runs, while random unlearning uses different data splits per seed.

### 4.3. Unlearning Efficacy and Utility Preservation

**Class Unlearning Results**  To evaluate this setting, we report **Forget Accuracy** ($A_f$) and **Retain Accuracy** ($A_r$). We explicitly evaluate $A_r$ on the *test set* because an entire class is removed; thus, preserving generalization to unseen samples of the remaining classes is the primary utility objective. We also report the **Test Cross-Entropy (CE)** relative to the retraining baseline. Table 1 highlights the superior scalability of our framework. While performance is saturated on CIFAR-10, distinct failure modes emerge on more complex datasets. In the **Standard Setting**, exact methods like SISA achieve perfect unlearning (e.g., $0.0\%$ $A_f$ on Tiny ImageNet) but are fundamentally limited by severe retraining overheads. Meanwhile, non-exact approximations often degrade utility (e.g., $67.2\%$ $A_r$ on CIFAR-100 with Bad Teacher; $48.9\%$ $A_r$ on Tiny ImageNet with SCRUB). In contrast, Rep. Unl. scales robustly, achieving near-perfect erasure while maintaining peak retention and the lowest Test CE. This advantage becomes especially pronounced in the **Zero-Shot Setting**, where baseline methods either fail to forget (e.g., B. Shrinking and GKT) or destabilize the model entirely (e.g., EMMN drops to $1.2\%$ $A_r$ on CIFAR-100). Rep. Unl. ZS reverses this trend by leveraging Neural Collapse proxies to maintain high utility and low Test CE, effectively anchoring the semantic manifold without requiring access to real retain data.

**Random Unlearning Results**  In this setting, standard forget accuracy is inapplicable, as forget samples belong to retained classes. Instead, we assess privacy using **Robust Membership Inference Attack (RMIA)** accuracy (Shokri et al., 2017; Zarifzadeh et al., 2023), where 0.5 indicates ideal unlearning. Due to train–test distribution shifts, RMIA can exceed 0.5 when forget samples resemble retain data, as both originate from the training set. We therefore consider the score *closest to the Retraining baseline* to be the most reliable privacy-utility target. Unlike class unlearning, we evaluate **Retain Accuracy** ($A_r$) on the *training set*. Because

*Table 1.* **Class Unlearning Results. Test** $A_r$ denotes accuracy on the retain classes (higher is better). **Test** $A_f$ denotes accuracy on the forget classes (lower is better). **Test CE** denotes the Cross-Entropy with the Retrain baseline, where lower values indicate better preservation of general model utility (lower is better).

| Method | CIFAR-10 | | | CIFAR-100 | | | Tiny ImageNet | | |
|---|---|---|---|---|---|---|---|---|---|
| | Test $A_r \uparrow$ | Test $A_f \downarrow$ | Test CE $\downarrow$ | Test $A_r \uparrow$ | Test $A_f \downarrow$ | Test CE $\downarrow$ | Test $A_r \uparrow$ | Test $A_f \downarrow$ | Test CE $\downarrow$ |
| *Reference Baselines* | | | | | | | | | |
| Pretrained | $93.1 \pm 0.1$ | $94.4 \pm 0.5$ | $1.97 \pm 0$ | $73.8 \pm 0.2$ | $71.2 \pm 0.7$ | $1.31 \pm 0$ | $\mathbf{59.7 \pm 0.4}$ | $64.9 \pm 1.0$ | $2.31 \pm 1$ |
| Retraining | $\mathbf{94.1 \pm 0}$ | $\mathbf{0.0 \pm 0}$ | $\mathbf{0.00 \pm 0}$ | $\mathbf{74.4 \pm 0}$ | $1.1 \pm 1$ | $\mathbf{0.00 \pm 0}$ | $59.7 \pm 0$ | $0.0 \pm 0$ | $0.00 \pm 0$ |
| Fine-tuning | $91.3 \pm 0$ | $12.9 \pm 4$ | $0.47 \pm 0$ | $69.0 \pm 1$ | $5.9 \pm 3$ | $1.59 \pm 0$ | $55.8 \pm 0$ | $7.6 \pm 2$ | $2.50 \pm 1$ |
| *Standard Setting (Access to $\mathcal{D}_r$ and $\mathcal{D}_f$)* | | | | | | | | | |
| SISA | $88.7 \pm 0$ | $\mathbf{0.0 \pm 0}$ | $0.44 \pm 0$ | $66.8 \pm 0$ | $\mathbf{0.0 \pm 0}$ | $1.58 \pm 0$ | $50.2 \pm 0$ | $\mathbf{0.0 \pm 0}$ | $3.05 \pm 0$ |
| SCRUB | $93.1 \pm 0$ | $\mathbf{0.0 \pm 0}$ | $0.31 \pm 0$ | $73.3 \pm 0$ | $9.9 \pm 3$ | $\mathbf{0.79 \pm 0}$ | $48.9 \pm 1$ | $\mathbf{0.0 \pm 0}$ | $2.11 \pm 1$ |
| UNSIR | $91.3 \pm 0$ | $14.1 \pm 11$ | $0.49 \pm 0$ | $72.4 \pm 0$ | $18.9 \pm 4$ | $0.86 \pm 0$ | $46.9 \pm 1$ | $46.3 \pm 2$ | $4.91 \pm 1$ |
| Bad Teacher | $92.8 \pm 0$ | $8.4 \pm 1$ | $0.52 \pm 0$ | $67.2 \pm 1$ | $10.7 \pm 1$ | $1.69 \pm 0$ | $56.0 \pm 0$ | $8.4 \pm 1$ | $2.50 \pm 1$ |
| Langevin | $62.0 \pm 0$ | $\mathbf{0.0 \pm 0}$ | $1.28 \pm 0$ | $32.0 \pm 0$ | $0.2 \pm 0$ | $2.93 \pm 0$ | $14.0 \pm 0$ | $0.2 \pm 0$ | $4.08 \pm 0$ |
| AMUN | $93.0 \pm 0$ | $\mathbf{0.0 \pm 0}$ | $0.35 \pm 0$ | $64.8 \pm 0$ | $44.8 \pm 0$ | $2.93 \pm 0$ | $49.9 \pm 0$ | $42.8 \pm 0$ | $4.40 \pm 0$ |
| NatMU | $92.6 \pm 0$ | $\mathbf{0.0 \pm 0}$ | $0.31 \pm 0$ | $71.0 \pm 0$ | $35.6 \pm 0$ | $1.40 \pm 0$ | $57.2 \pm 0$ | $42.2 \pm 0$ | $2.26 \pm 0$ |
| **Rep. Unl.** | $\mathbf{93.5 \pm 0}$ | $1.9 \pm 2$ | $0.36 \pm 0$ | $\mathbf{73.4 \pm 0}$ | $0.2 \pm 0$ | $0.88 \pm 0$ | $\mathbf{58.7 \pm 0}$ | $0.6 \pm 1$ | $1.80 \pm 1$ |
| *Zero-Shot Setting (No access to $\mathcal{D}_r$)* | | | | | | | | | |
| B. Shrinking | $83.7 \pm 1$ | $11.8 \pm 1$ | $1.86 \pm 0$ | $52.5 \pm 3$ | $20.7 \pm 2$ | $3.69 \pm 0$ | $47.9 \pm 1$ | $22.0 \pm 1$ | $4.17 \pm 1$ |
| EMMN | $23.3 \pm 3$ | $\mathbf{0.0 \pm 0}$ | $8.51 \pm 1$ | $1.2 \pm 0$ | $\mathbf{0.0 \pm 0}$ | $18.78 \pm 2$ | $0.5 \pm 0$ | $\mathbf{0.0 \pm 0}$ | $21.77 \pm 1$ |
| GKT | $46.6 \pm 24$ | $9.0 \pm 13$ | $2.41 \pm 1$ | $51.2 \pm 33$ | $43.6 \pm 29$ | $1.21 \pm 0$ | $45.5 \pm 2$ | $34.0 \pm 7$ | $2.20 \pm 0$ |
| **Rep. Unl. ZS** | $\mathbf{93.2 \pm 0}$ | $4.0 \pm 3$ | $\mathbf{0.42 \pm 0}$ | $\mathbf{69.3 \pm 0}$ | $0.2 \pm 0$ | $\mathbf{0.97 \pm 0}$ | $\mathbf{51.6 \pm 1}$ | $0.5 \pm 0$ | $1.98 \pm 0$ |

*Table 2.* **Random Data Unlearning Results. Train** $A_r$ ($\uparrow$) denotes accuracy on the retain set (higher is better). **MIA** denotes Membership Inference Attack accuracy, serving as a proxy for privacy risk; values closer to the Retraining baseline indicate successful unlearning. **Test CE** ($\downarrow$) denotes the Cross-Entropy with the Retrain baseline (lower is better). **Note:** UNSIR is excluded as it does not support random unlearning.

| Method | CIFAR-10 | | | CIFAR-100 | | | Tiny ImageNet | | |
|---|---|---|---|---|---|---|---|---|---|
| | Train $A_r \uparrow$ | RMIA | Test CE $\downarrow$ | Train $A_r \uparrow$ | RMIA | Test CE $\downarrow$ | Train $A_r \uparrow$ | RMIA | Test CE $\downarrow$ |
| Pretrained | $\mathbf{100.0 \pm 0}$ | $58.0 \pm 0.1$ | $0.48 \pm 0$ | $\mathbf{100.0 \pm 0}$ | $70.5 \pm 0.8$ | $1.07 \pm 0$ | $\mathbf{100.0 \pm 0}$ | $79.2 \pm 0.1$ | $1.72 \pm 0$ |
| Retraining | $\mathbf{100.0 \pm 0}$ | $\mathbf{54.2 \pm 0}$ | $\mathbf{0.00 \pm 0}$ | $\mathbf{100.0 \pm 0}$ | $68.7 \pm 1$ | $\mathbf{0.00 \pm 0}$ | $\mathbf{100.0 \pm 0}$ | $73.0 \pm 0$ | $\mathbf{0.00 \pm 0}$ |
| Fine-tuning | $98.2 \pm 0$ | $54.4 \pm 0$ | $0.31 \pm 0$ | $98.5 \pm 0$ | $64.1 \pm 1$ | $1.43 \pm 0$ | $97.8 \pm 1$ | $71.3 \pm 0$ | $2.15 \pm 0$ |
| SISA | $96.7 \pm 0$ | $49.9 \pm 0$ | $0.41 \pm 0$ | $87.2 \pm 0$ | $50.4 \pm 0$ | $1.84 \pm 0$ | $91.3 \pm 0$ | $49.9 \pm 0$ | $2.88 \pm 0$ |
| SCRUB | $99.9 \pm 0$ | $56.4 \pm 0$ | $0.18 \pm 0$ | $\mathbf{100.0 \pm 0}$ | $69.7 \pm 1$ | $0.80 \pm 0$ | $54.5 \pm 2$ | $51.2 \pm 0$ | $1.25 \pm 0$ |
| Bad Teacher | $88.7 \pm 1$ | $\mathbf{53.2 \pm 0}$ | $0.85 \pm 0$ | $71.5 \pm 1$ | $60.7 \pm 0$ | $3.28 \pm 0$ | $78.3 \pm 2$ | $69.8 \pm 0$ | $3.71 \pm 0$ |
| Langevin | $97.9 \pm 0$ | $55.1 \pm 0$ | $0.42 \pm 0$ | $31.3 \pm 0$ | $51.6 \pm 0$ | $2.92 \pm 0$ | $14.8 \pm 0$ | $51.2 \pm 0$ | $3.92 \pm 0$ |
| AMUN | $99.9 \pm 0$ | $55.6 \pm 0$ | $0.16 \pm 0$ | $\mathbf{100.0 \pm 0}$ | $75.6 \pm 0$ | $0.86 \pm 0$ | $\mathbf{100.0 \pm 0}$ | $80.5 \pm 0$ | $1.25 \pm 0$ |
| NatMU | $99.7 \pm 0$ | $55.1 \pm 0$ | $0.22 \pm 0$ | $99.4 \pm 0$ | $72.3 \pm 0$ | $1.38 \pm 0$ | $99.5 \pm 0$ | $78.6 \pm 0$ | $2.14 \pm 0$ |
| **Rep. Unl.** | $\mathbf{100.0 \pm 0}$ | $57.7 \pm 0$ | $\mathbf{0.14 \pm 0}$ | $99.9 \pm 0$ | $70.9 \pm 0$ | $\mathbf{0.69 \pm 0}$ | $\mathbf{100.0 \pm 0}$ | $75.6 \pm 0$ | $1.13 \pm 0$ |
| B. Shrinking | $95.5 \pm 1$ | $\mathbf{54.9 \pm 0}$ | $0.51 \pm 0$ | $29.9 \pm 1$ | $54.7 \pm 0$ | $7.68 \pm 0$ | $55.0 \pm 4$ | $62.9 \pm 1$ | $5.60 \pm 0$ |
| EMMN | $15.0 \pm 3$ | $50.1 \pm 0$ | $11.50 \pm 3$ | $1.2 \pm 0$ | $49.4 \pm 0$ | $18.73 \pm 1$ | $0.5 \pm 0$ | $49.4 \pm 1$ | $19.01 \pm 5$ |
| GKT | $41.3 \pm 24$ | $50.2 \pm 1$ | $2.55 \pm 1$ | $56.0 \pm 50$ | $58.0 \pm 7$ | $9.27 \pm 11$ | $63.3 \pm 5$ | $61.2 \pm 1$ | $2.05 \pm 0$ |
| **Rep. Unl. ZS** | $\mathbf{99.2 \pm 0}$ | $55.2 \pm 0$ | $0.45 \pm 0$ | $\mathbf{99.9 \pm 0}$ | $70.3 \pm 0$ | $0.81 \pm 0$ | $\mathbf{98.5 \pm 0}$ | $74.3 \pm 0$ | $1.69 \pm 0$ |

the random unlearning splits share the same class distribution, evaluating on the training set better isolates the model's ability to preserve specifically memorized retain instances. We also report **Test CE** to assess overall generalization. Table 2 shows that in the **Standard Setting**, baselines like Bad Teacher and SISA achieve marginally better RMIA at the cost of model integrity (high Test CE), while SCRUB again collapses on Tiny ImageNet ($54.5\% A_r$). Rep. Unl. prioritizes stability, consistently achieving near-perfect re-

tention and the lowest Test CE. Its slightly conservative RMIA tightly mirrors the Retraining baseline, ensuring a safer high-utility operating point. In the **Zero-Shot Setting**, the performance gap widens: B. Shrinking and EMMN fail on complex tasks (e.g., EMMN drops to $1.2\% A_r$). Rep. Unl. ZS stands out by achieving near-perfect retention, Retraining-level RMIA, and low Test CE—demonstrating that our proxies enable precise sample removal without access to retain data.

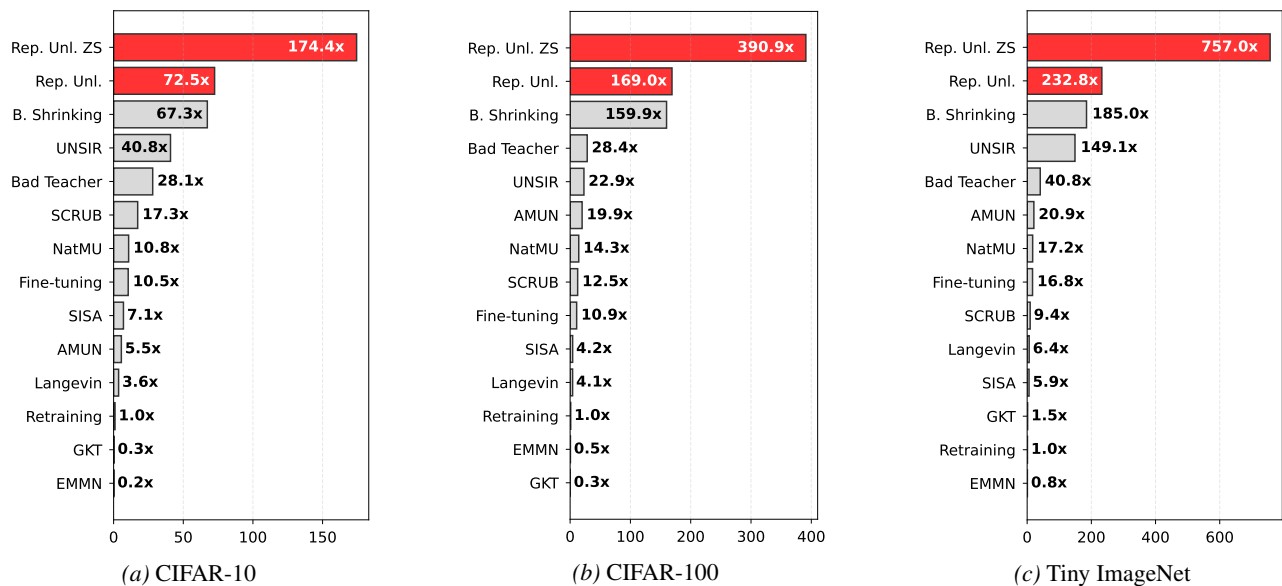

*Figure 4.* **Evaluation of Computational Efficiency.** A comparison of speed-up relative to full retraining for class unlearning on the CIFAR-10, CIFAR-100, and Tiny ImageNet datasets. Representation Learning methods are highlighted in red.

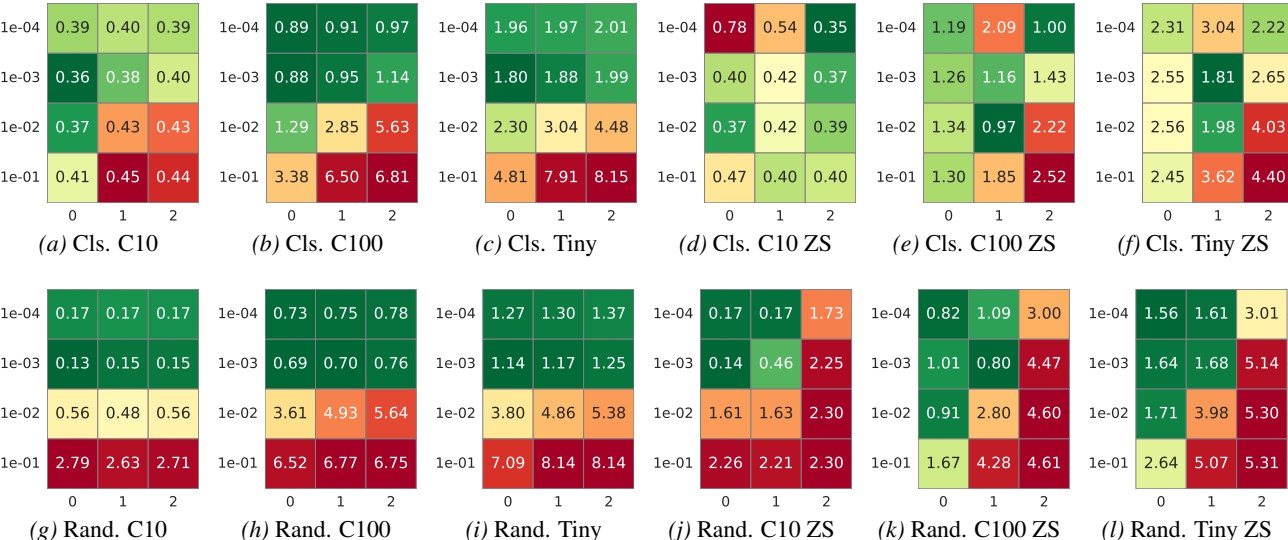

*Figure 5.* **Hyperparameter Sensitivity Analysis.** Effect of regularization strength $\beta$ (y-axis) and transformation depth (x-axis, number of hidden layers) on distributional fidelity. Values denote **Test Cross-Entropy (CE)** relative to the Retrain baseline, where lower values indicate better preservation of the original model's probability landscape.

### 4.4. Computational Efficiency.

Figure 4 reports the unlearning speed-ups relative to full retraining for the class unlearning scenario. Our framework demonstrates superior scalability, with efficiency gains increasing dramatically alongside dataset complexity. On the largest benchmark, Tiny ImageNet, Rep. Unl. ZS achieves a massive **757×** **speed-up**, significantly outpacing even our standard Rep. Unl. (233×). This internal advantage arises because the zero-shot variant processes strictly the forget set, avoiding the computational overhead of iterating through the retain data. Crucially, however, both variants outper-

form all competing baselines by orders of magnitude. This dominance is consistent across CIFAR-100 and CIFAR-10, where our two approaches consistently top the efficiency rankings. We observe analogous speed-ups for the random data unlearning protocol. Furthermore, beyond execution time, our method is highly resource-efficient; as detailed in Appendix E, we report the peak GPU memory allocation for all methods, showing that our approach minimizes memory overhead. This combination of speed and low resource consumption establishes our framework as the most viable candidate for large-scale unlearning deployments.

## 4.5. Hyperparameter Sensitivity Analysis

This section analyzes the sensitivity of our framework to the regularization strength $\beta$ and architectural complexity (hidden layers). We evaluate these effects using Test Cross-Entropy (CE) relative to the retrain baseline, as it captures global distributional fidelity. For an explicit joint analysis of privacy-utility trade-offs, we strongly direct readers to Appendix F.

**Impact of $\beta$**  We observe a consistent optimal performance around $\beta = 10^{-3}$. However, the **Standard Setting** is notably more sensitive to deviations from this value than the **Zero-Shot Setting**. In the standard scenario, excessive regularization leads to a sharp decline in distributional fidelity. In contrast, the zero-shot variant exhibits a more gradual degradation in utility, indicating a higher tolerance to manifold distortion when the model is decoupled from retain-data constraints.

**Impact of Hidden Layers**  The robustness of our method varies with data availability. In the **Standard Setting**, performance remains stable across architectures with 0, 1, or 2 hidden layers, suggesting that the required transformation is relatively simple. This observation aligns with the linear representation hypothesis and supports the use of lightweight architectures. In contrast, the **Zero-Shot Setting** achieves the best performance with a 1-hidden-layer transformation. We hypothesize that a purely linear mapping lacks sufficient capacity to reshape the representation space effectively, while a deeper (2-layer) network overfits due to limited supervision from the forget samples alone.

## 5. Limitations

While *Representation Unlearning* offers significant advantages in efficiency and stability, we acknowledge several practical and theoretical limitations.

First, as noted in Section 1, our method operates strictly under a black-box (API-access) threat model. Because the encoder parameters remain unchanged, the raw representation $Z$ still encodes information about the forget set, leaving it vulnerable to white-box recovery attacks. Our approach secures the exposed output $Z'$ via downstream intervention—akin to representation engineering—rather than expunging information from the raw weights.

Second, the introduction of the transformation layer $f_\phi$ imposes a permanent inference-time overhead. However, empirical profiling confirms this penalty is strictly on the microsecond scale (e.g., $0.59\,\mu$s for a 1-layer MLP on CIFAR-10, $1.09\,\mu$s on Tiny ImageNet). Consequently, a practitioner would need to perform billions of forward passes before the cumulative inference overhead exceeds the one-time cost of parameter-based retraining.

Third, the zero-shot variant relies heavily on the Neural Collapse hypothesis. If the pre-trained representation does not exhibit strong collapse, the quality of the class-conditional proxies degrades. This establishes an intrinsic upper bound on zero-shot retain utility, as classifier weights act as imperfect proxies for empirical class centers. We detail this vulnerability and the method's graceful degradation in Appendix G.

## 6. Future Directions

Future work could apply our same information-theoretic loss functions to fine-tune the entire encoder rather than learning a separate transformation $f_\phi$. This approach would simultaneously solve the first two limitations explained in the previous section: it achieves strict white-box erasure from the raw weights and eliminates the permanent inference-time overhead. However, we must note that modifying the full parameter space would significantly increase the computational cost of the unlearning process.

Furthermore, while this foundational study focuses on image classification, extending representation-level unlearning to Large Language Models (LLMs) is a highly promising direction. Inspired by recent advances in Representation Engineering (RepE) (Zou et al., 2023), applying our information-compression framework to the latent spaces of large foundation models could provide a scalable solution for unlearning complex behaviors or sensitive data without the profound instability and cost of updating billions of parameters.

## 7. Conclusion

In this work, we introduced a novel framework for machine unlearning that operates directly in the representation space. By replacing costly full-parameter updates with a lightweight feature transformation, our method circumvents the scalability bottlenecks and high GPU demands associated with weight-centric approaches. This modular design enables the targeted removal of information while preserving the backbone's semantic structure, achieving a superior stability–plasticity trade-off without relying on second-order information.

Comprehensive experiments on CIFAR-10, CIFAR-100, and Tiny ImageNet demonstrate that Representation Unlearning achieves state-of-the-art performance. In the standard setting, it delivers near-perfect forgetting and high utility even on complex datasets where parameter-based methods struggle. Notably, both variants offer substantial speed-ups; our Zero-Shot method achieves up to $757\times$ acceleration, eliminates the need for retain data, and overcomes the catastrophic collapse observed in prior baselines. This combination of efficiency, robustness, and scalability positions our framework as a practical solution for large-scale unlearning.

## Acknowledgements

This work has received funding from MCIN/AEI/10.13039/501100011033 under Grant PID2024-155948OB-C53.

## Impact Statement

This paper presents work whose goal is to advance the field of Machine Learning. There are many potential societal consequences of our work, none which we feel must be specifically highlighted here.

## Code Availability

An implementation accompanying this work is available at https://github.com/antonioalmudevar/representation_unlearning.

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

# A. Mathematical Derivations

### A.1. Derivation of Equation 2

We aim to upper bound the mutual information between the latent variables $Z$ and $Z'$ conditioned on the input $X_r$, denoted as $I(Z'; Z \mid X_r)$. We assume the Markov chain $X_r \to Z \to Z'$, which implies $p_\phi(z' \mid z, x_r) = p_\phi(z' \mid z)$.

To derive the bound, we introduce a variational approximation $r_\theta(z' \mid x_r)$ to the true marginal posterior $p_{\theta,\phi}(z' \mid x_r)$.

$$
\begin{aligned}
I(Z'; Z \mid X_r) &= \iiint p_{\theta,\phi}(z', z, x_r) \log \frac{p_\phi(z' \mid z)}{p_{\theta,\phi}(z' \mid x_r)} \, dz' \, dz \, dx_r \\
&= \iiint p(x_r) p_\theta(z \mid x_r) p_\phi(z' \mid z) \log \left( \frac{p_\phi(z' \mid z)}{r_\theta(z' \mid x_r)} \cdot \frac{r_\theta(z' \mid x_r)}{p_{\theta,\phi}(z' \mid x_r)} \right) \, dz' \, dz \, dx_r \\
&\qquad \text{(Multiply and divide by the variational distribution } r_\theta(z' \mid x_r)) \\
&= \mathbb{E}_{p_\theta(z,x_r)} \left[ \int p_\phi(z' \mid z) \log \frac{p_\phi(z' \mid z)}{r_\theta(z' \mid x_r)} \, dz' \right] + \mathbb{E}_{p(x_r)} \left[ \int p_{\theta,\phi}(z' \mid x_r) \log \frac{r_\theta(z' \mid x_r)}{p_{\theta,\phi}(z' \mid x_r)} \, dz' \right] \\
&\qquad \text{(Split the logarithm and identify expectations)} \\
&= \mathbb{E}_{p(x_r)} \left[ \mathbb{E}_{p_\theta(z \mid x_r)} \left[ D_{\mathrm{KL}}(p_\phi(z' \mid z) \| r_\theta(z' \mid x_r)) \right] \right] - \mathbb{E}_{p(x_r)} \left[ D_{\mathrm{KL}}(p_{\theta,\phi}(z' \mid x_r) \| r_\theta(z' \mid x_r)) \right] \quad (22)
\end{aligned}
$$

Since the Kullback-Leibler divergence is non-negative ($D_{\mathrm{KL}} \geq 0$), dropping the second term yields the following upper bound:

$$
I(Z'; Z \mid X_r) \leq \mathbb{E}_{p(x_r)} \left[ \mathbb{E}_{p_\theta(z \mid x_r)} \left[ D_{\mathrm{KL}}(p_\phi(z' \mid z) \| r_\theta(z' \mid x_r)) \right] \right] \quad (23)
$$

where the variational distribution is parameterized as a Gaussian, $r_\theta(z' \mid x) = \mathcal{N}(e_\theta(x), \sigma^2 I)$.

### A.2. Derivation of Equation 6

Here, we extend the bound by introducing label information $Y_r$. We assume the data distribution includes labels such that we integrate over the joint $p_{\theta,\phi}(z', z, x_r, y_r)$. We introduce a label-conditional variational approximation $r_\theta(z' \mid y_r)$.

$$
\begin{aligned}
I(Z'; Z \mid X_r) &= \iiiint p_{\theta,\phi}(z', z, x_r, y_r) \log \frac{p_\phi(z' \mid z)}{p_{\theta,\phi}(z' \mid x_r)} \, dz' \, dz \, dx_r \, dy_r \\
&= \mathbb{E}_{p_{\theta,\phi}(z',z,x_r,y_r)} \left[ \log \left( \frac{p_\phi(z' \mid z)}{r_\theta(z' \mid y_r)} \cdot \frac{r_\theta(z' \mid y_r)}{p_{\theta,\phi}(z' \mid x_r)} \right) \right] \\
&\qquad \text{(Introduce } r_\theta(z' \mid y_r) \text{ inside the logarithm)} \\
&= \mathbb{E}_{p(y_r)} \mathbb{E}_{p_\theta(z \mid y_r)} \int p_\phi(z' \mid z) \log \frac{p_\phi(z' \mid z)}{r_\theta(z' \mid y_r)} \, dz' \\
&\quad - \mathbb{E}_{p(x_r, y_r)} \int p_{\theta,\phi}(z' \mid x_r) \log \frac{p_{\theta,\phi}(z' \mid x_r)}{r_\theta(z' \mid y_r)} \, dz' \\
&\qquad \text{(Split terms and group into KL divergences)} \\
&= \mathbb{E}_{p(y_r)} \left[ \mathbb{E}_{p_\theta(z \mid y_r)} \left[ D_{\mathrm{KL}}(p_\phi(z' \mid z) \| r_\theta(z' \mid y_r)) \right] \right] \\
&\quad - \mathbb{E}_{p(x_r, y_r)} \left[ D_{\mathrm{KL}}(p_{\theta,\phi}(z' \mid x_r) \| r_\theta(z' \mid y_r)) \right] \quad (24)
\end{aligned}
$$

Utilizing the non-negativity of the KL divergence again, we obtain:

$$
I(Z'; Z \mid X_r) \leq \mathbb{E}_{p(y_r)} \left[ \mathbb{E}_{p_\theta(z \mid y_r)} \left[ D_{\mathrm{KL}}(p_\phi(z' \mid z) \| r_\theta(z' \mid y_r)) \right] \right] \quad (25)
$$

where the variational distribution is conditioned on the class label $c$, such that $r_\theta(z' \mid y_r = c) = \mathcal{N}(w_c, \sigma^2 I)$.

### A.3. Derivation of Equation 12

We wish to show that $I(Z'; X_f) \leq \mathbb{E}_{p(x_f)}[D_{\text{KL}}(p_{\theta,\phi}(z' \mid x_f) \parallel r_\theta(z'))]$. By definition, the Mutual Information is given by:

$$I(Z'; X_f) = \iint p(z', x_f) \log \frac{p_{\theta,\phi}(z' \mid x_f)}{r_{\theta,\phi}(z')} \, dz' \, dx_f$$

We introduce a variational marginal distribution $r_\theta(z')$ to approximate $r_{\theta,\phi}(z')$. We multiply and divide the term inside the logarithm by $r_\theta(z')$:

$$I(Z'; X_f) = \iint p(z', x_f) \log \left( \frac{p_{\theta,\phi}(z' \mid x_f)}{r_\theta(z')} \cdot \frac{r_\theta(z')}{r_{\theta,\phi}(z')} \right) dz' \, dx_f$$
$$= \iint p(z', x_f) \log \frac{p_{\theta,\phi}(z' \mid x_f)}{r_\theta(z')} \, dz' \, dx_f + \iint p(z', x_f) \log \frac{r_\theta(z')}{r_{\theta,\phi}(z')} \, dz' \, dx_f$$

The first term corresponds to the expected KL divergence conditioned on $x_f$. The second term simplifies to a negative KL divergence:

$$\text{Term 1} = \mathbb{E}_{p(x_f)} \left[ \int p_{\theta,\phi}(z' \mid x_f) \log \frac{p_{\theta,\phi}(z' \mid x_f)}{r_\theta(z')} \, dz' \right] = \mathbb{E}_{p(x_f)}[D_{\text{KL}}(p_{\theta,\phi}(z' \mid x_f) \parallel r_\theta(z'))]$$
$$\text{Term 2} = \int r_{\theta,\phi}(z') \log \frac{r_\theta(z')}{r_{\theta,\phi}(z')} \, dz' = -D_{\text{KL}}(r_{\theta,\phi}(z') \parallel r_\theta(z'))$$

Since the KL divergence is always non-negative, $D_{\text{KL}}(\cdot \parallel \cdot) \geq 0$, the second term is less than or equal to zero. Therefore:

$$I(Z'; X_f) \leq \mathbb{E}_{p(x_f)}[D_{\text{KL}}(p_{\theta,\phi}(z' \mid x_f) \parallel r_\theta(z'))]$$

### A.4. Derivation of Equation 15

First, we express the marginal distributions using their integral definitions:

- $p_{\theta,\phi}(z' \mid x_f) = \int p_\phi(z' \mid z) p_\theta(z \mid x_f) \, dz$
- $r_\theta(z') = \int r_\theta(z' \mid x) p(x) \, dx$

Substituting these definitions into the KL divergence, we apply Jensen's inequality relying on the joint convexity of the KL divergence. Since the divergence of the expectations is less than or equal to the expectation of the divergences, we obtain the following upper bound:

$$D_{\text{KL}}(p_{\theta,\phi}(z' \mid x_f) \parallel r_\theta(z')) = D_{\text{KL}} \left( \int p_\phi(z' \mid z) p_\theta(z \mid x_f) \, dz \,\middle\|\, \int r_\theta(z' \mid x) p(x) \, dx \right)$$
$$\leq \mathbb{E}_{p_\theta(z \mid x_f)} \left[ \mathbb{E}_{p(x)} \left[ D_{\text{KL}}(p_\phi(z' \mid z) \parallel r_\theta(z' \mid x)) \right] \right]$$

### A.5. Derivation of Equation 19

Analogous to the process in the previous derivation:

- $r_\theta(z') = \int r_\theta(z' \mid y) p(y) \, dy$

$$D_{\text{KL}}(p_{\theta,\phi}(z' \mid x_f) \parallel r_\theta(z')) = D_{\text{KL}} \left( \int p_\phi(z' \mid z) p_\theta(z \mid x_f) \, dz \,\middle\|\, \int r_\theta(z' \mid y) p(y) \, dy \right)$$
$$\leq \mathbb{E}_{p_\theta(z \mid x_f)} \left[ \mathbb{E}_{p(y)} \left[ D_{\text{KL}}(p_\phi(z' \mid z) \parallel r_\theta(z' \mid y)) \right] \right]$$

# B. Representation Unlearning Algorithms

---

**Algorithm 1** Standard Representation Unlearning

---

**Require:** Pre-trained encoder $e_\theta$, Forget set $\mathcal{D}_f$, Retain set $\mathcal{D}_r$, Transformation $f_\phi$.
**Require:** Hyperparameters: Learning rate $\eta$, Trade-off $\beta$
1: Initialize $\phi$ (e.g., identity initialization)
2: **while** not converged **do**
3:    **# 1. Sample Minibatches**
4:    Sample batch $x_f \sim \mathcal{D}_f$ and $x_r \sim \mathcal{D}_r$
5:    Sample reference batch $x \sim (\mathcal{D}_r \cup \mathcal{D}_f)$
6:    **# 2. Compute Latent Representations**
7:    $z_f \leftarrow e_\theta(x_f), \quad z_r \leftarrow e_\theta(x_r), \quad z \leftarrow e_\theta(x)$
8:    **# 3. Compute Retention Loss (Eq. 5)**
9:    $\mathcal{L}_r \leftarrow \frac{1}{B_r} \sum_i ||z_r^{(i)} - f_\phi(z_r^{(i)})||_2^2$
10:   **# 4. Compute Forget Loss (Eq. 17)**
11:   $\mathcal{L}_f \leftarrow \frac{1}{B_f B_{\text{ref}}} \sum_{i,j} ||z^{(j)} - f_\phi(z_f^{(i)})||_2^2$
12:   **# 5. Optimization Step**
13:   $\phi \leftarrow \phi - \eta \nabla_\phi (\mathcal{L}_r + \beta \mathcal{L}_f)$
14: **end while**
15: **return** Unlearned parameters $\phi^*$

---

**Algorithm 2** Zero-Shot Representation Unlearning

---

**Require:** Pre-trained encoder $e_\theta$, Forget set $\mathcal{D}_f$, Classifier Weights $W$, Transformation $f_\phi$.
**Require:** Hyperparameters: Learning rate $\eta$, Trade-off $\beta$.
**Require:** Metadata: Class counts $\{N^c\}_{c=1}^C$ (to estimate priors).
1: Extract class prototypes $w_c$ from $W$ (Neural Collapse).
2: Calculate class priors $p(y = c) = N^c/N$ and $p(y_r = c) = N_r^c/N_r$.
3: **while** not converged **do**
4:    **# 1. Sample Forget Data**
5:    Sample batch $x_f \sim \mathcal{D}_f$
6:    **# 2. Compute Latent Representations**
7:    $z_f \leftarrow e_\theta(x_f)$
8:    **# 3. Compute Zero-Shot Retention Loss (Eq. 10)**
9:    $\mathcal{L}_r^{zs} \leftarrow \sum_{c \in \mathcal{Y}_r} p(y_r = c)||w_c - f_\phi(w_c)||_2^2$
10:   **# 4. Compute Zero-Shot Forget Loss (Eq. 20)**
11:   $\mathcal{L}_f^{zs} \leftarrow \frac{1}{B_f} \sum_{i=1}^{B_f} \sum_{c=1}^C p(y = c)||w_c - f_\phi(z_f^{(i)})||_2^2$
12:   **# 5. Optimization Step**
13:   $\phi \leftarrow \phi - \eta \nabla_\phi (\mathcal{L}_r^{zs} + \beta \mathcal{L}_f^{zs})$
14: **end while**
15: **return** Unlearned parameters $\phi^*$

---

## C. Toy Dataset Details

We use a synthetic toy classification dataset generated from a $C$-component isotropic Gaussian mixture in $\mathbb{R}^d$, with $C = 6$ classes and $d = 10$ features. Classes are balanced, i.e., $p(y = c) = 1/C$ for $c \in \{0, 1, 2, 3, 4, 5\}$. For each class $c$, define $\theta_c = 2\pi c/C$ and a class mean $\mu_c \in \mathbb{R}^d$ whose first two coordinates lie on a circle of radius $R = 5$:

$$(\mu_{c,1}, \mu_{c,2}) = (R\cos\theta_c, \; R\sin\theta_c).$$

The remaining coordinates are sampled once per class as $\mu_{c,j} \sim \mathcal{N}(0, \tau^2)$ for $j = 3, \ldots, d$ with $\tau = 0.5$. Conditioned on $y = c$, examples are drawn i.i.d. as

$$x \mid (y = c) \sim \mathcal{N}(\mu_c, \sigma^2 I_d),$$

with $\sigma = 1$. Unless otherwise stated, we sample $n = 250$ points per class (total $N = Cn = 1500$). Training and test sets are generated independently by using different random seeds.

## D. Competing Hyperparameters

*Table 3.* Hyperparameters for Retraining

| Hyperparameter | C10 (C) | C10 (R) | C100 (C) | C100 (R) | TinyIM (C) | TinyIM (R) |
|---|---|---|---|---|---|---|
| Number of epochs | 100 | 100 | 200 | 200 | 160 | 160 |
| Learning rate | 0.001 | 0.001 | 0.001 | 0.001 | 0.001 | 0.001 |
| Max norm | 0 | 0 | 0 | 0 | 0 | 0 |
| Optimizer | adamw | adamw | adamw | adamw | adamw | adamw |
| Optimizer betas | [0.9, 0.999] | [0.9, 0.999] | [0.9, 0.999] | [0.9, 0.999] | [0.9, 0.999] | [0.9, 0.999] |
| Scheduler | cosine | cosine | cosine | cosine | cosine | cosine |
| Warmup epochs | 0 | 0 | 0 | 0 | 0 | 0 |
| Weight decay | 0.0005 | 0.0005 | 0.0005 | 0.0005 | 0.0005 | 0.0005 |

*Table 4.* Hyperparameters for Fine-tuning

| Hyperparameter | C10 (C) | C10 (R) | C100 (C) | C100 (R) | TinyIM (C) | TinyIM (R) |
|---|---|---|---|---|---|---|
| Number of epochs | 10 | 10 | 10 | 10 | 10 | 10 |
| Learning rate | 0.001 | 0.001 | 0.001 | 0.001 | 0.001 | 0.001 |
| Max norm | 0 | 0 | 0 | 0 | 0 | 0 |
| Optimizer | adam | adam | adam | adam | adam | adam |
| Weight decay | 0.0005 | 0.0005 | 0.0005 | 0.0005 | 0.0005 | 0.0005 |

*Table 5.* Hyperparameters for SISA

| Hyperparameter | C10 (C) | C10 (R) | C100 (C) | C100 (R) | TinyIM (C) | TinyIM (R) |
|---|---|---|---|---|---|---|
| Aggregation | logits | logits | logits | logits | logits | logits |
| Batch size | 128 | 128 | 128 | 128 | 128 | 128 |
| Epochs per slice | 2 | 2 | 2 | 2 | 1 | 1 |
| Learning rate | 0.1 | 0.1 | 0.1 | 0.1 | 0.1 | 0.1 |
| Momentum | 0.9 | 0.9 | 0.9 | 0.9 | 0.9 | 0.9 |
| Shards | 10 | 10 | 10 | 10 | 20 | 20 |
| Slices | 5 | 5 | 5 | 5 | 10 | 10 |
| Weight decay | 0.0005 | 0.0005 | 0.0005 | 0.0005 | 0.0005 | 0.0005 |

*Table 6.* Hyperparameters for SCRUB

| Hyperparameter | C10 (C) | C10 (R) | C100 (C) | C100 (R) | TinyIM (C) | TinyIM (R) |
|---|---|---|---|---|---|---|
| Alpha | 1 | 1 | 0.5 | 0.5 | 0.5 | 0.5 |
| Batch size forget | 512 | 512 | 256 | 256 | 512 | 512 |
| Batch size retain | 128 | 128 | 128 | 128 | 128 | 128 |
| Clip grad norm | 1 | 1 | 1 | 1 | 1 | 1 |
| Final min steps | 0 | 0 | 1 | 1 | 10 | 10 |
| Gamma | 1 | 1 | 1 | 1 | 1 | 1 |
| Learning rate | 0.0005 | 0.0005 | 0.0005 | 0.0005 | 0.0005 | 0.0005 |
| Lr decay after | 2 | 2 | 2 | 2 | 2 | 2 |
| Max steps | 2 | 2 | 5 | 5 | 1 | 1 |
| Min steps | 3 | 3 | 4 | 4 | 15 | 15 |
| Momentum | 0.9 | 0.9 | 0.9 | 0.9 | 0.9 | 0.9 |
| Optimizer | adam | adam | adam | adam | adam | adam |
| Rewind | false | false | false | false | false | false |
| Weight decay | 0.0005 | 0.0005 | 0.0005 | 0.0005 | 0.0005 | 0.0005 |

*Table 7.* Hyperparameters for UNSIR

| Hyperparameter | C10 (C) | C100 (C) | TinyIM (C) |
|---|---|---|---|
| Impair epochs | 1 | 3 | 8 |
| Impair LR | 0.02 | 0.05 | 0.003 |
| Max norm | 0 | 0 | 0 |
| Momentum | 0.9 | 0.9 | 0.9 |
| Noise batch size | 32 | 32 | 16 |
| Noise clamp | 2.5 | 2.5 | 2 |
| Noise copies | 20 | 20 | 15 |
| Noise iters | 25 | 25 | 20 |
| Noise l2 lambda | 0.1 | 0.1 | 0.2 |
| Noise LR | 0.01 | 0.01 | 0.01 |
| Repair epochs | 1 | 1 | 8 |
| Repair lr | 0.01 | 0.01 | 0.008 |
| Samples per retain class | 1000 | 1000 | 450 |
| Weight decay | 0 | 0 | 0 |

*Table 8.* Hyperparameters for Bad Teacher

| Hyperparameter | C10 (C) | C10 (R) | C100 (C) | C100 (R) | TinyIM (C) | TinyIM (R) |
|---|---|---|---|---|---|---|
| Number of epochs | 3 | 3 | 5 | 5 | 5 | 5 |
| Learning rate | 0.0005 | 0.0005 | 0.0005 | 0.0005 | 0.0005 | 0.0005 |
| Max norm | 0 | 0 | 0 | 0 | 0 | 0 |
| Momentum | 0.9 | 0.9 | 0.9 | 0.9 | 0.9 | 0.9 |
| Optimizer | sgd | sgd | sgd | sgd | sgd | sgd |
| Retain fraction | 0.3 | 0.3 | 0.3 | 0.3 | 0.3 | 0.3 |
| Temperature | 1 | 1 | 1 | 1 | 1 | 1 |
| Weight decay | 0.0005 | 0.0005 | 0.0005 | 0.0005 | 0.0005 | 0.0005 |

*Table 9.* Hyperparameters for Boundary Shrinking

| Hyperparameter | C10 (C) | C10 (R) | C100 (C) | C100 (R) | TinyIM (C) | TinyIM (R) |
|---|---|---|---|---|---|---|
| Batch size | 128 | 128 | 128 | 128 | 128 | 128 |
| Clamp max | 1 | 1 | 1 | 1 | 1 | 1 |
| Clamp min | 0 | 0 | 0 | 0 | 0 | 0 |
| Eps | 0.25 | 0.25 | 0.05 | 0.05 | 0.05 | 0.05 |
| Ft epochs | 10 | 10 | 10 | 10 | 15 | 15 |
| Ft lr | 1e-05 | 1e-05 | 1e-05 | 1e-05 | 1e-05 | 1e-05 |
| Ft momentum | 0.9 | 0.9 | 0.9 | 0.9 | 0.9 | 0.9 |
| Ft weight decay | 0 | 0 | 0 | 0 | 0 | 0 |
| Max norm | 0 | 0 | 0 | 0 | 0 | 0 |

*Table 10.* Hyperparameters for EMMN

| Hyperparameter | C10 (C) | C10 (R) | C100 (C) | C100 (R) | TinyIM (C) | TinyIM (R) |
|---|---|---|---|---|---|---|
| Impair LR | 0.01 | 0.01 | 0.01 | 0.01 | 0.01 | 0.01 |
| Impair steps | 2 | 2 | 2 | 2 | 2 | 2 |
| $\lambda_{reg}$ | 0.01 | 0.01 | 0.01 | 0.01 | 0.01 | 0.01 |
| Max norm | 0 | 0 | 0 | 0 | 0 | 0 |
| Noise batch size | 256 | 256 | 256 | 256 | 256 | 256 |
| Noise LR | 0.1 | 0.1 | 0.1 | 0.1 | 0.1 | 0.1 |
| Noise LR decay | 0.5 | 0.5 | 0.5 | 0.5 | 0.5 | 0.5 |
| Noise patience | 50 | 50 | 50 | 50 | 50 | 50 |
| Noise steps | 400 | 400 | 400 | 400 | 400 | 400 |
| Weight decay | 0.0001 | 0.0001 | 0.0001 | 0.0001 | 0.0001 | 0.0001 |

*Table 11.* Hyperparameters for GKT

| Hyperparameter | C10 (C) | C10 (R) | C100 (C) | C100 (R) | TinyIM (C) | TinyIM (R) |
|---|---|---|---|---|---|---|
| At beta | 250 | 250 | 250 | 250 | 250 | 250 |
| Batch size | 256 | 256 | 256 | 256 | 256 | 256 |
| Kl temperature | 1 | 1 | 1 | 1 | 1 | 1 |
| Learning rate | 0.001 | 0.001 | 0.001 | 0.001 | 0.001 | 0.001 |
| N generator iter | 1 | 1 | 1 | 1 | 1 | 1 |
| N pseudo batches | 4000 | 4000 | 4000 | 4000 | 4000 | 4000 |
| N student iter | 10 | 10 | 10 | 10 | 10 | 10 |
| Threshold | 0.01 | 0.01 | 0.01 | 0.01 | 0.01 | 0.01 |
| Z dim | 128 | 128 | 128 | 128 | 128 | 128 |

# E. GPU Memory Analysis

In this section, we provide a detailed analysis of the peak GPU memory allocation required by different unlearning methods. We measured the maximum memory usage (in MB) during the unlearning process, presenting the results in increasing order of dataset complexity: CIFAR-10, CIFAR-100, and Tiny ImageNet. This metric is crucial for determining the feasibility of deploying unlearning algorithms on resource-constrained edge devices or within multi-tenant cloud environments where memory is a limiting factor.

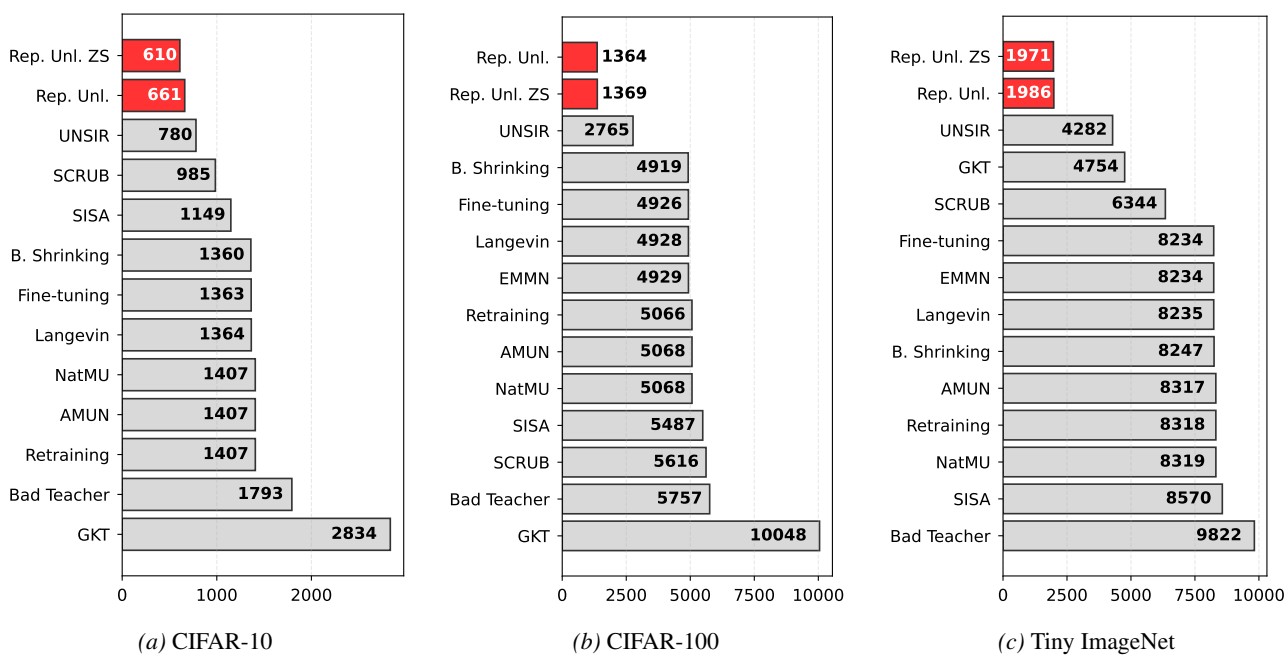

*(a)* CIFAR-10        *(b)* CIFAR-100        *(c)* Tiny ImageNet

*Figure 6.* **Peak GPU Memory Usage (MB).** Comparison of peak memory allocation across different methods. Our proposed frameworks (**Rep. Unl.** and **Rep. Unl. ZS**) consistently require the lowest memory footprint across all datasets, scaling efficiently from CIFAR-10 to Tiny ImageNet.

**CIFAR-10 Results** As shown in Figure 6a, our method demonstrates exceptional efficiency on the CIFAR-10 benchmark. **Rep. Unl. ZS** operates with a negligible peak allocation of **610 MB**, and **Rep. Unl.** follows closely at **661 MB**. This is significantly lower than standard **Retraining**, which consumes **1,407 MB**, and **GKT**, which requires **2,834 MB**—more than $4\times$ the memory of our zero-shot approach.

**CIFAR-100 Results** Moving to the more diverse CIFAR-100 dataset (Figure 6b), our framework maintains its resource advantage. **Rep. Unl.** requires approximately **1,364 MB**, while **Rep. Unl. ZS** requires **1,369 MB**. In contrast, we observe drastic memory spikes in baseline methods; **GKT** consumes a massive **10,048 MB**, and other methods such as SCRUB and SISA require between 5,000 and 5,600 MB.

**Tiny ImageNet Results** Finally, on the largest and most complex benchmark, Tiny ImageNet (Figure 6c), the scalability of our approach is most evident. **Rep. Unl. ZS** uses only **1,971 MB**, and **Rep. Unl.** uses **1,986 MB**. This stands in stark contrast to **Retraining**, which demands **8,318 MB**, and the **Bad Teacher** baseline, which reaches a peak of **9,822 MB**. Even the most efficient competing baseline, UNSIR, requires more than double the memory (4,282 MB) of our proposed framework.

**Discussion** The low memory footprint of our framework can be attributed to its design, which avoids the heavy computational graph overhead associated with end-to-end backpropagation on the full network. By operating primarily on the representation space and leveraging efficient closed-form or low-overhead updates, our approach decouples unlearning efficacy from high hardware demands. This confirms that our method is not only the fastest in terms of execution time but also the most strictly memory-efficient, making it uniquely suitable for scalable deployment.

# F. More Results on Ablation Studies

In this section, we provide a comprehensive sensitivity analysis of our unlearning framework. We evaluate the trade-off between unlearning efficacy (measured by accuracy on the forget set, $A_f$, or MIA AUC) and model utility (accuracy on the retain set, $A_r$). In the heatmaps presented below, the y-axis corresponds to the regularization hyperparameter $\beta \in \{10^{-4}, 10^{-3}, 10^{-2}, 10^{-1}\}$ (controlling the retain/forget trade-off), and the x-axis represents the number of hidden layers (0, 1, or 2) in the representation adapter $f_\theta$.

### F.1. Class Unlearning Protocol

For class unlearning, we aim to minimize $A_f$ while maintaining a high $A_r$. Figure 7 and Figure 8 illustrate these metrics across datasets.

**Impact on Retain Accuracy ($A_r$)**  We observe a distinct relationship between dataset complexity, adapter depth (x-axis), and regularization strength $\beta$ (y-axis).

*Robustness on Simple Datasets.* On CIFAR-10, retain accuracy is remarkably stable. In the standard (Non-ZS) protocol, accuracy remains above 90.9% even at maximal regularization ($\beta = 10^{-1}$), with negligible variance across depths. Similarly, the Zero-Shot (ZS) protocol maintains high utility ($> 93\%$) for depths 1 and 2, showing only a minor dip to 88.3% at depth 0 when $\beta = 10^{-1}$.

*Sensitivity in Complex Datasets.* For complex distributions (CIFAR-100, Tiny ImageNet), a "utility collapse" occurs when high regularization meets deeper adapters. In Non-ZS CIFAR-100, while shallow adapters (depth 0) retain 30.7% accuracy at $\beta = 10^{-1}$, deeper ones (depth 2) collapse to 1.0%. This trend intensifies on Tiny ImageNet, where accuracy drops to near-zero ($0.5\% - 0.8\%$) for depths 1 and 2 at $\beta = 10^{-1}$, suggesting deeper adapters overfit the unlearning objective at the expense of general representations.

*ZS Robustness.* The Zero-Shot variant is notably more resilient to this collapse. On Tiny ImageNet at $\beta = 10^{-1}$, the ZS protocol maintains 25.7% accuracy at depth 2, significantly outperforming the Non-ZS counterpart (0.5%).

*Conclusion.* While simple tasks allow aggressive unlearning (high $\beta$) regardless of adapter depth, complex tasks require a careful balance. Shallower adapters (depth 0 or 1) or moderate $\beta$ values ($\leq 10^{-2}$) are preferred to prevent catastrophic forgetting of the retain set.

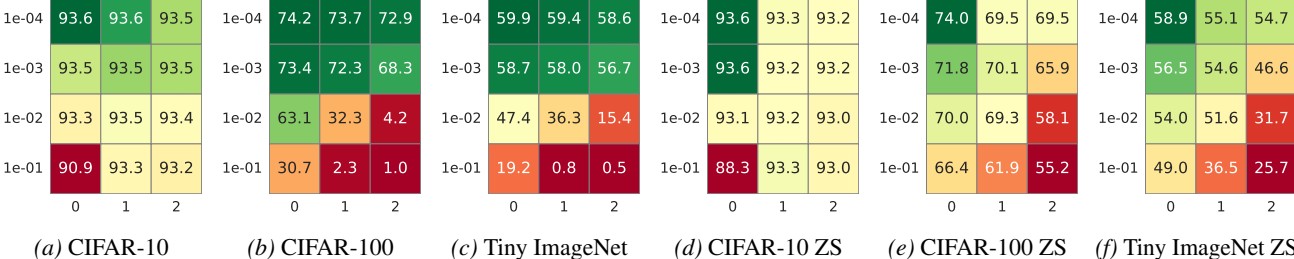

*Figure 7.* Test $A_r$ in *Class Unlearning*. Regularization strength $\beta$ (y-axis) and transformation depth (x-axis, number of hidden layers)

**Impact on Forget Accuracy ($A_f$)**  We observe a strong inverse correlation between regularization strength $\beta$ and forget accuracy $A_f$.

*Effect of Regularization.* Increasing $\beta$ consistently drives $A_f$ towards zero, ensuring effective unlearning. On Tiny ImageNet (Non-ZS), shifting $\beta$ from $10^{-4}$ to $10^{-2}$ reduces $A_f$ from $\sim 44.9\%$ to 0.0%, successfully erasing the target class. Similarly, on CIFAR-100 (Non-ZS), $A_f$ drops from $\sim 33.4\% - 37.0\%$ to near-zero levels as $\beta$ increases.

*Effect of Adapter Depth.* Deeper adapters demonstrate superior unlearning efficiency at lower regularization strengths. In the Zero-Shot (ZS) protocol for CIFAR-100 at $\beta = 10^{-4}$, a shallow adapter (depth 0) fails to unlearn ($A_f = 64.2\%$), whereas a depth-2 adapter achieves total forgetting without requiring strong regularization. This pattern repeats on Tiny ImageNet ZS, where depth-2 adapters reach 4.7% accuracy at $\beta = 10^{-4}$, compared to 49.8% for depth 0.

*Conclusion.* While high $\beta$ values ($\geq 10^{-1}$) guarantee unlearning across most configurations, they risk utility collapse. Deeper adapters offer a strategic advantage, enabling effective unlearning at lower $\beta$ values ($10^{-4} - 10^{-3}$), thereby preserving the stability of the retain set.

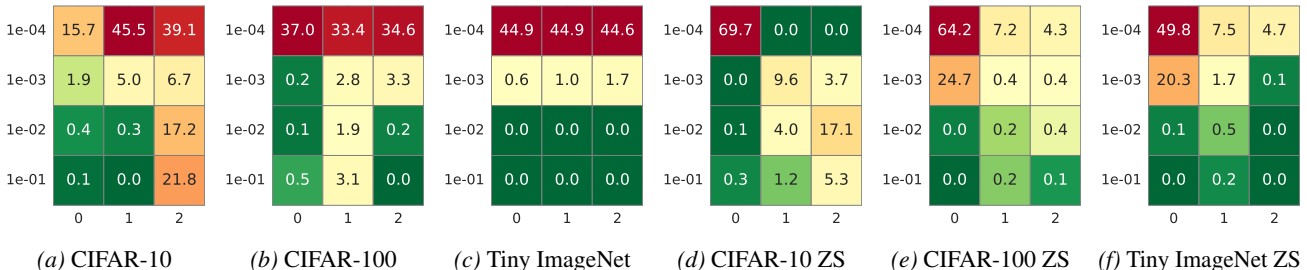

*Figure 8.* Test $A_f$ in *Class Unlearning*. Regularization strength $\beta$ (y-axis) and transformation depth (x-axis, number of hidden layers)

### F.2. Random Data Unlearning Protocol

In the random data unlearning scenario, we evaluate the method's ability to defend against membership inference attacks (MIA) while preserving utility.

**Impact on Retain Accuracy ($A_r$)** Random data unlearning exhibits significantly higher sensitivity to $\beta$ than class unlearning, creating a sharper trade-off between utility and forget-set erasure.

*Sensitivity and Collapse.* Across all datasets, setting $\beta$ too high ($\geq 10^{-2}$) risks a "utility collapse." For instance, on CIFAR-10 (Non-ZS), training accuracy remains perfect (100%) at $\beta \leq 10^{-3}$ but plummets to $\sim 12\% - 14\%$ at $\beta = 10^{-1}$. This threshold is even lower for complex datasets; on Tiny ImageNet (Non-ZS), the collapse begins at $\beta = 10^{-2}$, where accuracy drops to 46.8% (depth 0) and further to 5.7% (depth 2).

*The "Depth-0" Robustness in ZS.* A crucial observation in the Zero-Shot (ZS) protocol is the superior stability of shallow adapters. On Tiny ImageNet ZS, the depth-0 adapter is remarkably robust, maintaining 80.1% accuracy even at $\beta = 10^{-1}$, whereas the depth-2 adapter collapses completely to 1.1%. This trend is consistent on CIFAR-100 ZS, where depth-0 retains 83.9% at $\beta = 10^{-1}$ compared to just 2.8% for depth-2.

*Conclusion.* To preserve model utility in random data unlearning, one must operate within a stricter regularization regime ($\beta \leq 10^{-3}$) or leverage shallow adapters (depth 0) in the ZS setting, which offer a unique resilience against catastrophic forgetting.

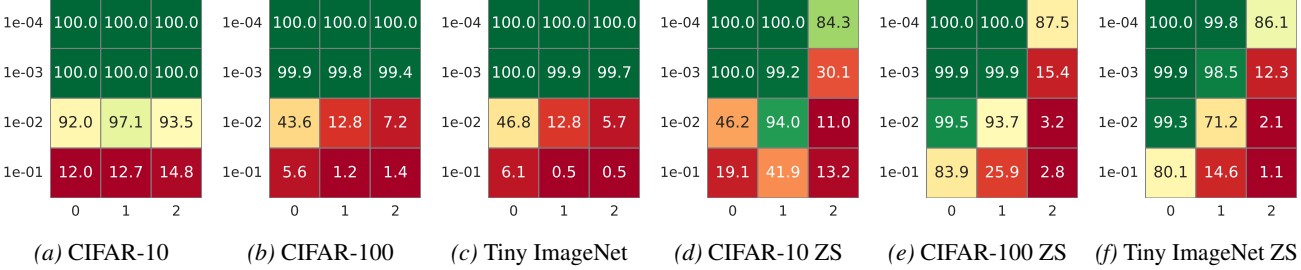

*Figure 9.* Train $A_r$ in *Random Data Unlearning*. Regularization strength $\beta$ (y-axis) and transformation depth (x-axis)

**MIA Efficacy and Privacy ($A_{mia}$)** We evaluate the defense against membership inference attacks, where an AUC of 50% represents ideal privacy (random guessing).

*Effect of Regularization.* Increasing $\beta$ consistently lowers MIA AUC, reducing information leakage. On CIFAR-10 (Non-ZS), the AUC drops from $\sim 57.8\%$ at $\beta = 10^{-4}$ to the ideal baseline of $\sim 50.0\%$ at $\beta = 10^{-1}$. This trend is even more dramatic on complex datasets like Tiny ImageNet (Non-ZS), where high regularization reduces the AUC from a vulnerable 81.7% to a safe 49.3%.

*Depth as a Privacy Enabler.* Deeper adapters are significantly more effective at scrubbing private information, particularly in the Zero-Shot (ZS) protocol. On Tiny ImageNet ZS, while the depth-0 adapter fails to defend against attacks (retaining a high AUC of 78.4% even at $\beta = 10^{-1}$), the depth-2 adapter successfully reduces the AUC to 50.3%. Similarly, on CIFAR-100 ZS, depth-2 adapters reach near-perfect privacy (50.2%) at moderate regularization ($\beta = 10^{-2}$), whereas depth-0 adapters remain vulnerable (73.7%).

*Conclusion.* To guarantee privacy against MIA, particularly in challenging Zero-Shot scenarios, deeper adapters (depth 2) are essential. However, this creates a trade-off: deeper adapters provide superior privacy but carry a higher risk of utility collapse (as discussed in Figure 9), necessitating careful hyperparameter tuning.

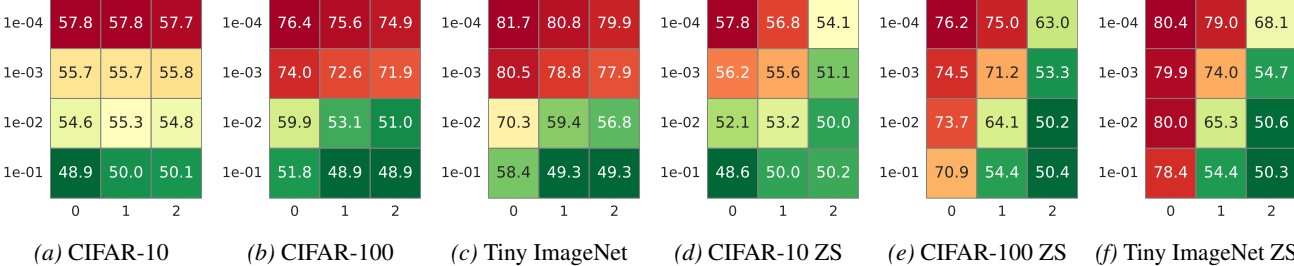

*(a)* CIFAR-10     *(b)* CIFAR-100     *(c)* Tiny ImageNet     *(d)* CIFAR-10 ZS     *(e)* CIFAR-100 ZS     *(f)* Tiny ImageNet ZS

*Figure 10.* MIA AUC in *Random Data Unlearning*. Regularization strength $\beta$ (y-axis) and transformation depth (x-axis)

## G. Neural Collapse Diagnostics and Zero-Shot Degradation

Our zero-shot formulation relies on the Neural Collapse (NC) hypothesis, specifically assuming that classifier weights can serve as reliable proxies for class-conditional feature centers. In this section, we empirically validate this hypothesis on our pre-trained models and evaluate the method's sensitivity to imperfect proxy conditions through a graceful degradation analysis.

### G.1. Baseline Neural Collapse Diagnostics

To quantify the geometric relationship between pre-trained features and classifier weights prior to unlearning, we evaluate the representations using the standard Neural Collapse framework. We compute the following diagnostics:

- **NC1:** Within-class collapse relative to between-class separation; lower indicates each class is better summarized by a single prototype.

- **NC3:** Alignment between classifier weights and empirical class means; higher indicates the weights better match class prototypes.

- **NC4:** Agreement between the learned classifier and the nearest-class-center rule; higher indicates the classifier behaves more like a prototype-based classifier.

- **Acc. Gap:** The difference between classifier and nearest-class-center training accuracy; lower indicates less utility is lost when replacing the classifier with class prototypes.

As shown in Table 12, the degree to which the pre-trained representation fulfills Neural Collapse serves as an intrinsic bottleneck for strictly zero-shot unlearning. On CIFAR-10, the proxy is nearly exact (NC4 = 0.9996, Acc. Gap = 0.0004), indicating that classifier weights behave almost identically to class prototypes. On CIFAR-100, this agreement remains strong but weakens slightly. On Tiny ImageNet, the proxy remains reasonably aligned but is less exact (NC4 = 0.9355), together with worse NC1, showing that the representation is less collapsed and therefore less well captured by a single prototype.

*Table 12.* **Baseline Neural Collapse Diagnostics.** Evaluating the geometric alignment between representations and classifier weights across the pre-trained models.

| Dataset | NC1 $\downarrow$ | NC3 $\uparrow$ | NC4 $\uparrow$ | Acc. Gap $\downarrow$ |
|---|---|---|---|---|
| CIFAR-10 | 0.2116 | 0.8188 | 0.9996 | 0.0004 |
| CIFAR-100 | 2.0760 | 0.7462 | 0.9908 | 0.0090 |
| Tiny ImageNet | 7.8577 | 0.7005 | 0.9355 | 0.0643 |

## G.2. Zero-Shot Graceful Degradation Analysis

Without access to retain examples, any residual mismatch between the classifier weight $w_y$ and the true class-conditional geometry defines an intrinsic upper bound on zero-shot retain utility. To test the method's behavior when proxy alignment is inherently poor, we artificially degrade the proxies by injecting increasing Gaussian noise ($\sigma \in \{0.0, 0.1, 0.2, 0.3\}$) into the target weights.

The results for class unlearning (Table 13) and random unlearning (Table 14) demonstrate that our method is sensitive to the Neural Collapse hypothesis holding true. As $\sigma$ increases and targets become less reliable, retain utility incurs a severe and proportional penalty (e.g., CIFAR-10 retain accuracy drops from 93.2% to 71.5% in class unlearning). However, crucially, *unlearning efficacy is not affected*. Forget Accuracy remains near 0%, and RMIA safely drops toward the random guess limit, maintaining strong privacy. This confirms that if proxy alignment is poor, the method still successfully erases targeted concepts, gracefully trading off retain utility rather than catastrophically failing to unlearn.

*Table 13.* **Zero-Shot Graceful Degradation Analysis (Class Unlearning).** Evaluating unlearning robustness when injecting noise ($\sigma$) into the proxy weights in the class unlearning scenario. **Test** $A_r$ ($\uparrow$) and **Test** $A_f$ ($\downarrow$) denote the Retain Test and Forget Test accuracies (%). **Test CE** ($\downarrow$) is the Cross-Entropy divergence. Unlearning efficacy (Test $A_f$) remains remarkably stable near zero even as utility degrades. Results report the mean $\pm$ standard deviation across 5 random seeds.

| Noise ($\sigma$) | CIFAR-10 | | | CIFAR-100 | | | Tiny ImageNet | | |
|---|---|---|---|---|---|---|---|---|---|
| | Test $A_r$ $\uparrow$ | Test $A_f$ $\downarrow$ | Test CE $\downarrow$ | Test $A_r$ $\uparrow$ | Test $A_f$ $\downarrow$ | Test CE $\downarrow$ | Test $A_r$ $\uparrow$ | Test $A_f$ $\downarrow$ | Test CE $\downarrow$ |
| 0.0 | $93.2 \pm 0.3$ | $6.4 \pm 3.9$ | $0.42 \pm 0.01$ | $70.1 \pm 0.1$ | $0.6 \pm 0.3$ | $1.15 \pm 0$ | $54.6 \pm 0.6$ | $1.1 \pm 0.3$ | $1.81 \pm 0.42$ |
| 0.1 | $91.9 \pm 0.6$ | $0.1 \pm 0.2$ | $0.36 \pm 0.03$ | $68.4 \pm 0.2$ | $0.4 \pm 0.3$ | $1.08 \pm 0.03$ | $52.9 \pm 0.5$ | $1.7 \pm 0.5$ | $1.79 \pm 0.32$ |
| 0.2 | $83.5 \pm 4.4$ | $0.8 \pm 1.6$ | $0.68 \pm 0.08$ | $64.3 \pm 0.6$ | $1.0 \pm 0.2$ | $1.15 \pm 0.04$ | $49.3 \pm 0.3$ | $3.5 \pm 1.1$ | $1.90 \pm 0.19$ |
| 0.3 | $71.5 \pm 8.6$ | $1.4 \pm 1.2$ | $1.18 \pm 0.11$ | $59.4 \pm 0.6$ | $1.7 \pm 0.9$ | $1.50 \pm 0.04$ | $44.2 \pm 0.2$ | $6.0 \pm 1.6$ | $2.39 \pm 0.09$ |

*Table 14.* **Zero-Shot Graceful Degradation Analysis (Random Unlearning).** Evaluating unlearning robustness when injecting noise ($\sigma$) into the proxy weights in the random unlearning scenario. **Train** $A_r$ ($\uparrow$) denotes the Retain Train accuracy (%). **RMIA** is the privacy metric (%); values closer to the 50% random guess limit indicate successful unlearning. **Test CE** ($\downarrow$) is the Cross-Entropy divergence. Results report the mean $\pm$ standard deviation across 5 random seeds.

| Noise ($\sigma$) | CIFAR-10 | | | CIFAR-100 | | | Tiny ImageNet | | |
|---|---|---|---|---|---|---|---|---|---|
| | Train $A_r$ $\uparrow$ | RMIA | Test CE $\downarrow$ | Train $A_r$ $\uparrow$ | RMIA | Test CE $\downarrow$ | Train $A_r$ $\uparrow$ | RMIA | Test CE $\downarrow$ |
| 0.0 | $98.8 \pm 0.5$ | $55.3 \pm 0.3$ | $0.47 \pm 0.05$ | $99.9 \pm 0$ | $70.6 \pm 0.3$ | $0.80 \pm 0.01$ | $98.5 \pm 0.1$ | $74.2 \pm 0.1$ | $1.69 \pm 0.04$ |
| 0.1 | $86.6 \pm 4.3$ | $52.1 \pm 0.3$ | $1.46 \pm 0.04$ | $99.6 \pm 0$ | $69.2 \pm 0.3$ | $1.06 \pm 0.01$ | $97.0 \pm 0.3$ | $72.3 \pm 0.2$ | $1.92 \pm 0.04$ |
| 0.2 | $48.3 \pm 5.6$ | $48.6 \pm 0.3$ | $2.02 \pm 0.03$ | $97.0 \pm 0.2$ | $65.1 \pm 0.2$ | $1.88 \pm 0.02$ | $90.7 \pm 0.4$ | $67.9 \pm 0.2$ | $2.58 \pm 0.03$ |
| 0.3 | $38.3 \pm 7.1$ | $48.2 \pm 0.5$ | $2.07 \pm 0.04$ | $89.6 \pm 0.8$ | $60.9 \pm 0.2$ | $2.74 \pm 0.02$ | $78.8 \pm 0.6$ | $62.8 \pm 0.3$ | $3.39 \pm 0.03$ |

