# OpenReview forum: "Representation Unlearning: Forgetting through Information Compression"
_ICML.cc/2026/Conference — ICML 2026 regular_

### Official Review · Reviewer_Qz1G · 2026-03-10

**Soundness:** 3
**Presentation:** 3
**Significance:** 3
**Originality:** 3
**Overall Recommendation:** 4
**Confidence:** 4

**Summary:**

This paper proposes Representation Unlearning, a framework for machine unlearning that operates in the model's representation space rather than the parameter space. The key idea is to learn a lightweight transformation $f_\phi$ over the last hidden layer $Z$, enforcing an information bottleneck: maximizing mutual information with retained data $\mathcal{D}_r$ while suppressing mutual information with forget data $\mathcal{D}_f$. The authors derive variational upper bounds for the resulting intractable mutual information terms and study the framework under two data-access regimes: a standard setting (access to both $\mathcal{D}_r$ and $\mathcal{D}_f$) and a zero-shot setting (access to $\mathcal{D}_f$ only), the latter leveraging Neural Collapse proxies to bypass the need for retain samples. Experiments on CIFAR-10, CIFAR-100, and Tiny ImageNet report competitive unlearning–utility trade-offs together with large computational speed-ups over retraining.

**Compliance With Llm Reviewing Policy:**

Affirmed.

**Final Justification:**

The reviewer have addressed most of my concerns in a satisfactory manner.

**Key Questions For Authors:**

1. Algorithm 2 vs. the derivation: prior mismatch. In the zero-shot retention loss (Eq. 10), the per-class weight is $N_c^r$ (the retain prior from Eq. 7), whereas Algorithm 2 (line 9) appears to use the global prior $p(y=c) = N_c/N$ (line 2). For class unlearning this distinction vanishes ($N_c^r = N_c$ for retained classes), but for random unlearning with uniform removal across classes, $N_c^r < N_c$ and the two priors disagree up to a non-trivial factor. Could you confirm whether this is a notational oversight or a deliberate simplification, and report any impact on random unlearning performance?
2. MIA implementation and the 67% Retraining baseline. The 67% MIA accuracy for Retraining on CIFAR-100 raises concerns about the validity of the attack. Could you provide full implementation details (number of shadow models, classifier architecture, data splits, training epochs) and justify why this baseline is reliable? Would using a stronger, instance-level MIA (e.g., LiRA from Carlini et al., 2022) change the conclusions? A clear answer here would significantly affect my confidence in the privacy evaluation, as the current instantiation is severely limited.
3. Zero-shot setting: known class repartition. The zero-shot variant assumes knowledge of the per-class counts ${N_c}_{c=1}^C$. Is this assumption standard in the zero-shot unlearning literature? Do competing zero-shot baselines (B. Shrinking, EMMN, GKT) also leverage this metadata? While not vital to the take-aways of the paper, this should be mentioned.
4. Eq. 17 intuition and the role of $p(x)$ in the forget loss. In Eq. 17, the updated representation of the forget set is matched to the original (pre-transformation) representation of the full training set. I do not discuss the mathematical soundness of this choice, but would appreciate the authors’ intuition on it. Intuitively, one might expect the target to be the post-transformation representation of the overall data, since nothing guarantees that the retain representations remain in place after $f_\phi$ is applied. Could you clarify the intuition behind this and discuss whether matching against post-transformation statistics ($f_\phi(z^{(j)})$ rather than $z^{(j)}$) would change the bound or the empirical results in you understanding.
5. Number of runs and class selection. How many independent runs were used to compute the standard deviations in Tables 1 and 2? Were different forget classes (or data splits) used across runs, or was the same configuration repeated with different seeds?

**Limitations:**

The discussion of limitations could be substantially improved. In particular:
* The permanent inference-time overhead introduced by $f_\phi$ is not discussed as a limitation, despite being a key practical trade-off compared to weight-based methods.
* The reliance on the Neural Collapse hypothesis in the zero-shot setting is acknowledged implicitly but not critically examined: for models or datasets where Neural Collapse does not hold well, the proxy quality may degrade.
* The MIA evaluation limitations (acknowledged briefly) deserve a more thorough treatment, given that the privacy guarantees of the method rest partly on this metric.

**Strengths And Weaknesses:**

Strengths
* Principled information-theoretic formulation. The use of mutual information objectives, combined with variational bounds and the information bottleneck perspective, provides a well-grounded theoretical basis for unlearning in representation space. The mathematical derivations appear sound and well-presented.
* Elegant zero-shot variant. Leveraging the Neural Collapse hypothesis to construct class-conditional proxies that replace retain-set access is a creative contribution. This is a practically important setting, and the zero-shot variant performs remarkably well relative to the baselines.
* Computational efficiency. The speed-ups over retraining and competing methods are substantial and well-documented. The low GPU memory footprint (Appendix E) further strengthens the practical appeal of the approach.
* Thorough hyperparameter sensitivity analysis. The paper includes a comprehensive grid over $\beta$ and architecture depth across all settings, which is informative and well-presented (Section 4.5 and Appendix F).
* Clear mechanistic illustration. The 2D toy experiment (Section 4.1) provides good intuition for how the transformation reshapes the representation space in both the standard and zero-shot settings.

Weaknesses
* Incomplete related work and missing baselines. The discussion of zero-shot methods is limited (~7 lines). More importantly, existing methods that already operate in the feature/representation space, such as (but not limited to) "Selective Unlearning via Representation Erasure Using Domain Adversarial Training" and "Deep Unlearning: Fast and Efficient Gradient-Free Class Forgetting" (the latter also avoiding gradients) are not discussed. The paper would benefit from positioning itself against these works and including experimental comparisons, or at the very least should discuss them. In addition, the "do nothing" baseline (equivalent to setting $\beta=0$ in the proposed method) is a natural reference point, more so even than the fine-tune baseline in my opinion, and its absence is a limiting factor.
* Weak MIA evaluation. The membership inference attack evaluation has several shortcomings. (i) Only Shokri et al. (2017) is referenced, but the MIA literature has advanced significantly since then, e.g., instance-level attacks (as opposed to population-level attacks in Shokri’s work), such as those proposed in "Membership Inference Attacks From First Principles" (Carlini et al., 2022). (ii) The specific formulation of MIA for unlearning has been discussed, e.g., in "Inexact Unlearning Needs More Careful Evaluations to Avoid a False Sense of Privacy", an important milestone in how to properly evaluate unlearning through MIAs, and is not acknowledged here. (iii) No experimental details are provided regarding the attack implementation: number of shadow models, training duration, data splits, classifier architecture, and number of epochs are all missing. (iv) The 67% and 68% MIA accuracy reported for Retraining on CIFAR-100 and Tiny ImageNet , even though briefly addressed by the authors, appears suspiciously high and may indicate an erroneous attack instantiation, which cannot be verified due to the lack of detail.
* Inconsistent and insufficiently justified metrics. In the class unlearning setting (Table 1), retain accuracy is measured on the test set ("test retain"), while in the random data unlearning setting (Table 2), it is measured on the training set. The paper mentions distribution shifts, but it is unclear why retain accuracy on the actual retain set cannot be reported in both cases. The term "test retain set" is itself non-standard and could be clarified. The use of consistent metrics across settings would strengthen the experimental narrative.
* Missing experimental details. (i) No information is provided about which class(es) are unlearned in the class unlearning setting. (ii) The number of runs used to compute standard deviations is not reported, and it is unclear whether different runs use different data splits or forget classes. (iii) $\omega_c$ is not formally introduced before use. (iv) Minor: missing period at l.182.
* Inference-time overhead not discussed. While the proposed method avoids expensive parameter-level updates, it introduces 1–2 additional layers ($f_\phi$) that incur a permanent per-sample overhead at every future inference. This trade-off relative to weight-based methods (which pay a higher upfront cost but no per-inference penalty) is not discussed. It would be valuable to characterize the break-even point: how many inferences can be performed before the cumulative overhead of representation unlearning exceeds the one-time cost of parameter-based unlearning?
* Concerns with SISA results. In Figure 4, SISA is reported to take $\times 1.0$ to $\times 1.4$ the time of retraining, suggesting near-full retraining. 10 or 20 shards are mentioned in the appendix. If all shards contain forget data, this explains the running time but not the 58.9% forget accuracy on the forgotten class of Tiny ImageNet. If only some shards are affected, there should be a speed-up. Either interpretation raises questions that are not resolved in the text.
* Sensitivity analysis only uses CE. While the $\beta$ sensitivity analysis (Section 4.5) is appreciated, evaluating only the cross-entropy loss is insufficient, especially for random data unlearning, where test utility is already well preserved without doing anything. The authors could use two joint metrics to find the sweet spot for $\beta$, or one metric that explicitly balances utility and privacy, such as the one proposed in « Evaluation Metrics for the NeurIPS Machine Unlearning Competition », as it would be more informative.

---

> ### Author Rebuttal · Authors · 2026-03-29
>
> We sincerely thank the reviewer for their constructive and detailed feedback. We are particularly glad you found our information-theoretic formulation principled, the zero-shot variant elegant, and our hyperparameter analysis thorough. To address your review efficiently, we have organized our response into four thematic blocks. Additionally, we will add a dedicated "Limitations" section to the revised manuscript to formally address the constraints discussed below.
>
> Anonymous PDF with new tables/figures: https://osf.io/d5tjf/files/gd4x6?view_only=e4ca7deea0ce481caee83fa60c44f900
>
> ---
> ## **Block 1: Upgrading the Privacy Evaluation (W2, Q2, Lim 3)**
>
> We agree that outdated MIAs give a false sense of privacy. We have upgraded to a state-of-the-art, instance-level attack: the Robust Membership Inference Attack (RMIA). Please see our detailed response to **Reviewer jntj (Weakness 4)** for the full experimental setup and updated results. We will expand Section 4.3 to detail this SOTA evaluation.
>
> We were initially surprised by the >50% MIA as well, but verified our data splits are strictly disjoint. The elevated Retraining MIA arises naturally because the forget set is i.i.d. with the training data; the retrained model simply generalizes better to it ($\approx$98% accuracy) than to the test set ($\approx$93%) when trained only on the retain set. Therefore, the target metric is not an absolute AUC $\approx$ 50%, but rather AUC(unlearned) $\approx$ AUC(retrain).
>
> Regarding implementation, we used a standard loss-based MIA (and now RMIA) computed directly from the target model's cross-entropy loss, requiring no shadow models or extra hyperparameters.
>
> ---
> ## **Block 2: Theoretical & Algorithmic Choices**
>
> * **Algorithm 2 Typo (Q1):** This is a notation typo in the pseudocode. Our code correctly utilizes the class-conditional prior $p_k$. We will correct Algorithm 2.
> * **Eq. 17 Intuition (Q4):** While Eq. 5 encourages retain representations to remain static, $f_\phi$ is practically not a perfect identity function. Targeting $f_\phi(z)$ ensures the updated forget representations geometrically align with the newly transformed representation space, which the classification head actually uses to compute logits.
> * **Zero-shot Priors (Q3):** Competing baselines do not assume access to class counts $N_k$; we will note this. If $N_k$ is strictly unknown, substituting it with a uniform prior $p(c)$ maintains the variational bounds.
> * **Neural Collapse (Lim 2):** Addressed in our response to **Reviewer RoUx (Weakness 1)**.
>
> ---
> ## **Block 3: Computational Trade-offs (Overhead & SISA)**
>
> * **Inference-Time Overhead (W5, Lim 1):** We agree the added $f_\phi$ layer introduces a permanent inference overhead, which we will discuss in our Limitations. However, profiling confirms the absolute inference time penalty is strictly on the microsecond scale:
>
>   | Dataset | Linear (μs) | 1 Layer (μs) | 2 Layer (μs) |
>   | :--- | :--- | :--- | :--- |
>   | CIFAR-10 | 0.43 | 0.59 | 0.77 |
>   | CIFAR-100 | 2.86 | 3.99 | 4.78 |
>   | Tiny ImageNet | 0.49 | 0.87 | 1.09 |
>
>   To characterize the break-even point against parameter-based methods, we compare this per-sample penalty to the time of a single retraining run (1075s for CIFAR-10, 7715s for CIFAR-100, and 21,395s for Tiny ImageNet). Using the 1 Layer architecture (the largest model in our main results), a practitioner must perform ~1.8 billion inferences for CIFAR-10 ($1075/(0.59\times10^{-6})$), ~1.9 billion for CIFAR-100, and ~24.6 billion for Tiny ImageNet before the cumulative inference overhead reaches the one-time cost of retraining.
>
> * **SISA Results (W6):** We address this point in our response to **Question 3 of Reviewer 2YjE**.
>
> ---
> ## **Block 4: Experimental Details, Baselines, and Metrics**
>
> * **Missing Baselines (W1):** We will add the suggested baselines to Section 2. The "do nothing" baseline is now included in Tables 4 and 5 of the attached document.
> * **Inconsistent Metrics (W3):** Metric choice depends strictly on the task. In *class unlearning*, we forget entire classes, so we evaluate on the test set to verify retain class accuracy is not degraded. In *random unlearning*, we forget specific samples across all classes, meaning a "retain test accuracy" cannot be meaningfully calculated. Instead, we measure accuracy on the retain training set to verify the model has not lost its capacity to memorize the remaining training samples. This is vaguely discussed in line 326, but will be better discussed and clarified in the final version.
> * **Missing Details (W4, Q5):** We used 5 seeds for all standard deviations. Class unlearning forgot the exact same class across runs; random unlearning used different data splits. We will introduce all these details.
> * **Sensitivity Metric (W7):** Due to space limits, the explicit joint utility/privacy analysis is located in Appendix F. We will add a stronger pointer to it in the main text.

---

> > ### Author Rebuttal · Reviewer_Qz1G · 2026-04-01
> >
> > I thank the reviewers for their rebuttal. They have addressed most of my concern in a satisfactory manner. I am increasing my evaluation to a weak accept.

---

> > > ### Author Response · Authors · 2026-04-06
> > >
> > > We sincerely thank the reviewer for their time, constructive feedback, and active engagement during the rebuttal process. We are very glad that our response satisfactorily addressed your concerns, and we greatly appreciate your updated evaluation and support for our work.

---

### Official Review · Reviewer_2YjE · 2026-03-11

**Soundness:** 3
**Presentation:** 3
**Significance:** 2
**Originality:** 4
**Overall Recommendation:** 4
**Confidence:** 4

**Summary:**

This paper proposes Representation Unlearning, a post-hoc machine unlearning framework that operates in the feature space rather than modifying model parameters. The method learns a lightweight transformation fφ that maps penultimate-layer representations Z to Z′ by optimizing an information-bottleneck style objective: retain information about the remaining data while suppressing information about the forget set. The authors derive variational upper bounds that reduce to simple MSE losses under Gaussian parameterization and extend the approach to a zero-shot regime by leveraging Neural Collapse–based class prototypes, reporting strong forgetting/retention and large speed-ups on CIFAR-10/100 and Tiny ImageNet.

**Compliance With Llm Reviewing Policy:**

Affirmed.

**Final Justification:**

Mosf of issues have been solved

**Key Questions For Authors:**

1. Could you try applying this idea to image generation? There is no need for excessively large models or datasets; using the SD1.5 model and the Imagenette dataset, you should be able to generate results quite quickly.
2. How sensitive is the zero-shot variant to violations of Neural Collapse? Can you report quantitative NC diagnostics (e.g., within-class covariance, alignment of class means to weights) and correlate them with ZS performance?
3. The SISA results for class unlearning on Tiny ImageNet appear unusually weak. What is your precise SISA protocol for handling class deletions across shards/slices, and how many shards/slices were retrained per request?

**Limitations:**

yes

**Strengths And Weaknesses:**

Strengths:
1. Framing unlearning as information compression in representation space is an appealing alternative to parameter-space updates, with a principled objective grounded in mutual information and practical variational surrogates.
2. The zero-shot instantiation that uses Neural Collapse proxies (classifier weights) to approximate class-conditional feature distributions is simple, elegant, and avoids expensive data synthesis or teacher models.
3. The approach is modular and lightweight: a small linear or shallow MLP fφ applied to penultimate features is easy to train and deploy.
4. The problem setup, objectives, and variational bounds are clearly introduced, with derivations provided in appendices and concise pseudocode for both regimes.
Weakness:
1. Unlearning in representation space is novel. But it should be extend to other task, not only image classification, but also generation, graph, LLM tasks.
2. The zero-shot instantiation relies heavily on Neural Collapse and access to per-class counts; the degree to which NC holds in the evaluated settings (and beyond) is not quantified, and departures from NC could degrade behavior.

---

> ### Author Rebuttal · Authors · 2026-03-29
>
> We thank the reviewer for the detailed review and the constructive suggestions. We address your questions below.
>
> Anonymous PDF with new tables/figures: https://osf.io/d5tjf/files/gd4x6?view_only=e4ca7deea0ce481caee83fa60c44f900
>
> ---
> **Q1: Application to Image Generation**
>
> We thank the reviewer for this suggestion. Extending our representation unlearning methodology to generative models—including Diffusion Models, VAEs, and LLMs—is a highly promising direction that we actively plan to explore in future work. However, we believe that adapting and evaluating the method for the generative domain falls outside the scope of the present paper. Introducing the necessary generative evaluation protocols and baselines would risk distracting the reader from the core focus on classification models, and it would significantly exceed our current space limitations.
>
> ---
> **Q2: Sensitivity to Neural Collapse and NC Diagnostics**
>
> Please see our detailed response to **Reviewer RoUx (W1 & Q1: Quantifying Baseline NC Diagnostics)**. In that response, we provide a comprehensive empirical analysis addressing this exact concern. We quantify the baseline proxy alignment (reporting the mean alignment and variance) and present a zero-shot graceful degradation experiment to explicitly evaluate the method's sensitivity to violations of the Neural Collapse hypothesis.
>
> ---
> **Q3: SISA Implementation Flaws and Corrected Results**
>
> We thank the reviewer for identifying this issue (Reviewer Qz1G raised similar concerns). Upon careful inspection, we found two flaws in our previous SISA implementation that explain both the weak forget accuracy on Tiny ImageNet and the near-retrain running times.
>
> First, the shard retraining budget was insufficient — too many shards with too few epochs per slice meant that after removing forget data, the partial retraining could not fully overcome the memorization of the forget classes, resulting in the residual forget accuracy observed on Tiny ImageNet. Adjusting the number of shards and epochs per slice resolved this.
>
> Second, the training loop performed a full evaluation on the retain and forget sets after every epoch of every slice of every shard, which dominated the measured runtime and obscured the genuine speedup from partial shard retraining.
>
> With both flaws corrected, SISA achieves 0.0 forget accuracy across all settings (Table 4) and recovers the expected speedups of 7.1×, 4.2×, and 5.9× over retraining on CIFAR-10, CIFAR-100, and Tiny ImageNet respectively (Figure 4), as reported in the attached document.

---

> > ### Author Rebuttal · Reviewer_2YjE · 2026-04-03
> >
> > Image classification tasks is too simple. However, unlearning in representation holds immense potential. I will maintain my current level of support.

---

> > > ### Author Response · Authors · 2026-04-06
> > >
> > > We sincerely thank the reviewer for their continued support and for recognizing the immense potential of representation-level unlearning.
> > >
> > > Regarding your initial suggestion about generative models, we completely agree that scaling this approach to generative tasks is a highly exciting and relevant direction for future work. However, for this foundational study, we focused on image classification because it remains the standard testbed for evaluating unlearning methods. In fact, much of the existing literature relies on simpler settings, whereas our evaluation actively scales up to more complex benchmarks like CIFAR-100 and TinyImageNet.
> > >
> > > We hope our work serves as a stepping stone toward representation unlearning in generative contexts, and we greatly appreciate your time and positive assessment.

---

### Official Review · Reviewer_jntj · 2026-03-13

**Soundness:** 2
**Presentation:** 3
**Significance:** 2
**Originality:** 2
**Overall Recommendation:** 3
**Confidence:** 4

**Summary:**

This paper proposes a framework for machine unlearning that operates on the representation space instead of directly modifying model parameters. The key idea is to learn a transformation $f_\phi$ that maps the original latent representation $z$ to a new representation $z'$ that preserves information relevant to retained data while removing information associated with the forget set. The authors formulate the problem with an information-theoretic objective where the transformed representation should retain information about the retain set while minimizing mutual information with the forget set. The paper considers both a standard setting and a zero-shot setting with experiments on CIFAR-10, CIFAR-100, and Tiny ImageNet.

**Compliance With Llm Reviewing Policy:**

Affirmed.

**Final Justification:**

Although the experiments are through and show good results, they black-box setting assumed by the authors is not the realistic machine unlearning setting which mandates the removal of any trace of the data from the trained models as it keeps the encoder as it is.

**Key Questions For Authors:**

Please see weaknesses. I would be willing to raise my score if authors address the weaknesses.

**Limitations:**

Yes

**Strengths And Weaknesses:**

## Strengths

- The paper is generally well written and easy to follow.
- The motivation for operating in representation space rather than directly modifying model parameters is clearly stated.
- The proposed approach provides a practical alternative to parameter-based methods by learning lightweight representation transformations. The reported computational speedups relative to retraining are substantial.


---

## Weaknesses

- The proposed approach leaves the encoder parameters unchanged and only learns a transformation on top of the latent representation. As a result, the original representation $z$ produced by the encoder may still encode information about the forget set. It is unclear to me whether the method truly removes the influence of the forgotten data or instead masks it through a downstream transformation. It would be helpful to evaluate whether the forgotten information can still be recovered from the original representations using probing or reconstruction attacks.

- The paper argues that the influence of individual samples is more localized in representation space, which motivates editing representations instead of parameters. However, this claim lacks support from empirical analysis within the paper. Additional experiments like layer-wise analysis or probing experiments could strengthen this motivation.

- Although the paper frames the method as a new representation-level unlearning framework, related ideas have been explored in prior work on representation editing and concept erasure. For example, methods such as LEACE [1] and kernelized rate-distortion–based concept erasure [2] remove targeted information from representations through projection or representation transformations. The paper would benefit from a clearer discussion of how the proposed approach differs from or improves upon these related methods.

- The only privacy method used for evaluating the quality of unlearning is an outdated MIA. More recent MIAs have been shown to greatly outperform these earlier methods [3,4]. Recent unlearning methods have adapted these to the setting of unlearning evaluation ([5] uses the MIA in [4] and [7] uses the MIA in [3]). There are more recent works on stronger MIAs for unlearning evaluation [6]. Since the approximate methods, such as the one presented in this paper, do not come with theoretical guarantees for unlearning, they need to utilize SOTA evaluation methods for empirical evaluations. Otherwise, they give a “false sense of privacy” [6].

- Several recent approximate unlearning methods [7,10], as well as scalable certified ones [8,9] are missing in the evaluations which does not allow the reader to find out where this method stands in comparison to prior SOTA. Zero-shot setting is also a main focus in [7].

- The proposed objective minimizes variational bounds on mutual information quantities, but the paper does not provide guarantees that optimizing these objectives produces a model equivalent to retraining on the retain set. As a result, it is unclear whether the method satisfies any formal notion of machine unlearning beyond empirical performance (and therefore the empirical evaluations become more important).

[1] Belrose, N., Schneider-Joseph, D., Ravfogel, S., Cotterell, R., Raff, E., & Biderman, S. (2023). Leace: Perfect linear concept erasure in closed form. Advances in Neural Information Processing Systems, 36, 66044-66063.

[2] Basu Roy Chowdhury, S., Monath, N., Dubey, K. A., Ahmed, A., & Chaturvedi, S. (2023). Robust concept erasure via kernelized rate-distortion maximization. Advances in Neural Information Processing Systems, 36, 43284-43306.

[3] Zarifzadeh, S., Liu, P., and Shokri, R. Low-cost high-power membership inference attacks. In Forty-first International Conference on Machine Learning, 2024.

[4] Carlini, N., Chien, S., Nasr, M., Song, S., Terzis, A., & Tramer, F. (2022, May). Membership inference attacks from first principles. In 2022 IEEE symposium on security and privacy (SP) (pp. 1897-1914). IEEE.

[5] Kurmanji, M., Triantafillou, P., Hayes, J., & Triantafillou, E. (2023). Towards unbounded machine unlearning. Advances in neural information processing systems, 36, 1957-1987.

[6] Hayes, J., Shumailov, I., Triantafillou, E., Khalifa, A., & Papernot, N. (2025, April). Inexact unlearning needs more careful evaluations to avoid a false sense of privacy. In 2025 IEEE Conference on Secure and Trustworthy Machine Learning (SaTML) (pp. 497-519). IEEE.

[7] Ebrahimpour-Boroojeny, A., Sundaram, H., & Chandrasekaran, V. (2025, October). Not all wrong is bad: Using adversarial examples for unlearning. In Forty-second International Conference on Machine Learning.


[8] Chien, E., Wang, H., Chen, Z., & Li, P. (2024). Langevin unlearning: A new perspective of noisy gradient descent for machine unlearning. Advances in neural information processing systems, 37, 79666-79703.

[9] Zhang, B., Dong, Y., Wang, T., & Li, J. (2024). Towards certified unlearning for deep neural networks. arXiv preprint arXiv:2408.00920.

[10] He, Z., Li, T., Cheng, X., Huang, Z., & Huang, X. (2025). Towards natural machine unlearning. IEEE Transactions on Pattern Analysis and Machine Intelligence.

---

> ### Author Rebuttal · Authors · 2026-03-29
>
> We thank the reviewer for the careful and constructive feedback. We address their questions in detail below.
>
> Anonymous PDF with new tables/figures: https://osf.io/d5tjf/files/gd4x6?view_only=e4ca7deea0ce481caee83fa60c44f900
>
> ---
> **W1: Masking vs. Parameter-Level Unlearning**
>
> We agree: because the encoder parameters remain unchanged, the raw representation $Z$ still encodes the forget set and remains vulnerable to white-box recovery attacks.
>
> Thus, our method operates strictly under a black-box (API-access) threat model, where users interact only with the transformed output $Z'$. Akin to LLM activation engineering, we secure the exposed output via a downstream intervention rather than performing computationally prohibitive weight updates. We will clarify this threat model and the white-box vulnerability in a new "Limitations" section.
>
> ---
> **W2: Localized Influence in Representation Space**
>
> To validate that sample influence is localized in representation space ($Z$) but widely distributed in parameter space ($\theta$), we analyzed the absolute footprint of sample-specific gradients.
>
> **Experiment & Results:** For 500 CIFAR-10 training samples, we computed and sorted the absolute classification loss gradients with respect to parameters ($\nabla_\theta L$) and final representations ($\nabla_Z L$). We then calculated the number of dimensions required to capture a growing fraction (0.1 to 1.0) of the total gradient mass.
>
> As shown in the Cumulative Gradient Mass Curve (**Figure 1 in the document in the link**), the scale difference spans four orders of magnitude:
> * **Distributed Influence ($\theta$):** Capturing a meaningful gradient mass fraction requires aggregating hundreds of thousands to millions of parameter weights.
> * **Localized Influence ($Z$):** The exact same relative influence is highly concentrated, bounded entirely below the 512-dimensional capacity of $Z$.
>
> This confirms sample identity is broadly entangled in $\theta$ yet tightly isolated in $Z$. We will include this analysis in the appendix.
>
> ---
> **W3: Concept Erasure and Representation Editing**
>
> We thank the reviewer for highlighting these two works. We will add a Related Work subsection distinguishing our framework from concept erasure:
>
> * **Task:** Concept erasure globally removes a *specific, shared attribute* (e.g., protected traits) across a dataset. Machine Unlearning instead completely removes the *holistic identity and influence* of specific training instances or entire classes.
> * **Methodology:** LEACE [1] uses closed-form orthogonal projections, and kernelized RD [2] relies on kernel estimators requiring empirical data and concept labels. Conversely, we learn a lightweight transformation $f_\phi$ over representation space $Z$. As noted in W1, leaving the original encoder frozen trades exact parameter-level erasure for massive computational speedups.
> * **Zero-Shot Capability:** Our Information Bottleneck formulation yields tractable variational bounds exploiting Neural Collapse (NC). This enables zero-shot unlearning using classifier weights as proxies, bypassing the need for retain data—a capability absent in standard concept erasure.
>
> ---
> **W4. Upgraded Privacy Evaluation via SOTA MIAs**
>
> We agree stronger evaluations prevent a "false sense of privacy." We implemented the recommended RMIA (**Tables 4 and 5 in the linked PDF**). The results exhibit trends closely matching our original MIA, corroborating our initial conclusions and confirming our method's efficacy. The final manuscript will prominently feature RMIA instead of the traditional MIA.
>
> ---
> **W5. Comparison with Recent SOTA and Missing Baselines ([7]-[10])**
>
> We have implemented AMUN [7], Langevin Unlearning [8], and NatMU [10], and added them to **Tables 4 and 5 in the linked document**. We also attempted to evaluate [9] but faced persistent convergence issues.
>
> Our expanded empirical analysis yields two main conclusions regarding these recent methods:
> 1. **Langevin Unlearning [8]:** Struggles primarily with retain accuracy, scaling poorly as dataset and architecture complexity increase.
> 2. **AMUN [7] and NatMU [10]:** Struggle primarily with forget accuracy, facing similar scaling issues specifically in the class unlearning scenario.
>
> Because the original papers do not evaluate beyond CIFAR-10, these scaling degradation issues at higher complexities were previously unreported. We will include these baselines in the revision.
>
> ---
> **W6: Formal Guarantees**
>
> We agree. Because classical unlearning definitions are anchored in parameter space, establishing formal theoretical equivalence with our representation-based approach is inherently difficult. Acknowledging this gap, we rely on rigorous empirical validation. We have strengthened our experiments in the revision—most notably by including state-of-the-art RMIA evaluations—to concretely validate our method's privacy posture where traditional parameter-based theories fall short.

---

> > ### Author Rebuttal · Reviewer_jntj · 2026-04-04
> >
> > The authors have improved the empirical evaluation by adding stronger MIAs and additional baselines, and they provide useful analysis on Neural Collapse and representation locality. However, the core concern remains: the method does not perform true unlearning, as the encoder continues to encode information about the forget set. The authors also acknowledge this limitation. Thus, the method risks providing only superficial unlearning.
> >
> > The authors justify this by stating that they consider machine unlearning under a black-box threat model. However, this contradicts one of the main motivations of machine unlearning, "the right to be forgotten" granted to data owners. Upon such requests, the model owner is expected to remove the data from the model, rather than retain it in the encoder and merely alter the representation and final results.
> >
> > Still due to the thorough responses to my other concerns, I have raised my score to 3. However, the aforementioned limitation prevents me from increasing my score further.

---

> > > ### Author Response · Authors · 2026-04-06
> > >
> > > We thank the reviewer for acknowledging our empirical improvements and raising their score. Regarding concerns about true unlearning and the "right to be forgotten," we respectfully offer two perspectives:
> > >
> > > **1. The Complexity of Parameter Space vs. Representation Engineering (RepE)**
> > > While we agree that completely expunging information from the raw weights is the theoretical ideal, modifying the parameter space directly is notoriously complex, highly entangled, and computationally prohibitive as models scale. This is exactly why methods like Representation Engineering [1] have become increasingly central to the field, particularly for large foundation models. In practice, companies are recognizing representation-level interventions as the most efficient and stable method for guaranteeing behavioral modifications and safeguarding outputs under black-box scenarios, avoiding the instability of high-dimensional weight updates.
> > >
> > > **2. Threat Models and the "Right to be Forgotten"**
> > > Regarding the right to be forgotten, we respectfully note that all unlearning methods—even exact parameter-updating algorithms—fundamentally rely on a trust boundary with the model owner. If the threat model assumes a malicious owner, exact unlearning also fails, as the owner could simply retain a pre-unlearning checkpoint. Conversely, assuming a compliant model owner, legally mandating the strict application of our representation transformation provides the exact same functional guarantee to the end-user as mandating the deletion of old checkpoints. Both paradigms ultimately rely on compliance; ours simply achieves the necessary output-level forgetting with significantly less computational overhead.
> > >
> > >
> > > [1] Zou, A., Phan, L., Chen, S., Campbell, J., Guo, P., Ren, R., ... & Hendrycks, D. (2023). Representation engineering: A top-down approach to ai transparency. arXiv preprint arXiv:2310.01405.
> > >
> > > ---
> > >
> > > We deeply appreciate your rigorous evaluation, which has helped us better define the practical scope of our contribution.

---

### Official Review · Reviewer_RoUx · 2026-03-13

**Soundness:** 3
**Presentation:** 3
**Significance:** 3
**Originality:** 2
**Overall Recommendation:** 5
**Confidence:** 3

**Summary:**

This paper investigates machine unlearning from a representation-space viewpoint. The main idea is to perform forgetting by learning a transformation of latent representations. The proposed framework is derived from an information-theoretic objective that aims to suppress information associated with the forget data while preserving useful information for the retain data. The paper considers both a standard setting with access to retain and forget samples and a zero-shot setting that uses limited metadata together with classifier weights to approximate the retain objective. Experimental results suggest that the method can provide a favorable trade-off between forgetting effectiveness, retained utility, and computational efficiency.

**Compliance With Llm Reviewing Policy:**

Affirmed.

**Final Justification:**

This paper studies an machine unlearning problem from a representation space perspective. I find this viewpoint interesting and potentially useful, especially because it offers a lightweight alternative to more expensive retraining based approaches. The paper also considers both a standard setting and a zero shot setting, and the empirical results suggest that the proposed method can achieve a competitive balance between forgetting effectiveness, retained utility, and efficiency. Taking the paper and the rebuttal together, I believe the rebuttal addressed my main concerns and improved my understanding of the paper. Consequently, I am raising my overall recommendation to accept, and I have also increased my confidence score.

**Key Questions For Authors:**

- The zero shot formulation appears to rely critically on the assumption that classifier weights can serve as reliable proxies for class conditional feature centers. Could the authors provide either empirical evidence or a more detailed justification showing that this approximation is sufficiently accurate for the models and datasets considered in the paper. More broadly, how sensitive is the zero shot method to violations of the Neural Collapse style assumption. For example, if classifier weights are imperfect proxies for class conditional representation centers, does the method degrade gracefully, or is its performance highly dependent on this approximation being very accurate. A clearer analysis here would help assess the robustness and practical scope of the zero shot formulation.
- For the class unlearning setting, the paper evaluates forgetting mainly through forget class accuracy together with utility related metrics. Could the authors clarify whether they have additional evidence that more directly supports the claimed verifiability of unlearning, for example through privacy evaluations. This matters because low forget accuracy alone does not necessarily establish that the forgotten information is no longer identifiable or recoverable in a stronger sense.

**Limitations:**

yes

**Strengths And Weaknesses:**

Strengths
- The paper explores machine unlearning from a representation space viewpoint rather than the more common parameter update perspective. This angle is conceptually interesting and gives the work a clear methodological identity.
- By learning a transformation on latent representations of a fixed pretrained model, the method aims to suppress information related to the forget data while preserving useful information for the retain data. This design is intuitive and computationally appealing.
- The paper presents both a standard setting and a zero shot setting. The zero shot variant is particularly interesting because it attempts to perform unlearning with only limited metadata and classifier weights, which broadens the practical scope of the framework.
- The experimental results suggest that the proposed method can often achieve a favorable balance between forgetting effectiveness, retained utility, and efficiency.

Weaknesses
- The zero shot formulation relies on a strong Neural Collapse that uses classifier weights as proxies for class conditional feature centers. This assumption is central to the proposed retain objective in the zero shot setting, yet the paper does not provide direct empirical evidence that it is sufficiently accurate for the considered models and datasets.
- The empirical evaluation does not fully support the paper’s stronger claims regarding verifiability. For class unlearning, the evidence is based mainly on forget class accuracy and related utility metrics. These are useful indicators, but they are weaker than direct privacy evaluations.

---

> ### Author Rebuttal · Authors · 2026-03-29
>
> We thank the reviewer for the careful and constructive feedback. We address your questions in detail below.
>
> Anonymous PDF with new tables/figures: https://osf.io/d5tjf/files/gd4x6?view_only=e4ca7deea0ce481caee83fa60c44f900
>
> ---
> **W1 & Q1: Quantifying Baseline NC Diagnostics**
> *(Note: We refer Reviewers 2YjE and Qz1G to this discussion, as it addresses related concerns regarding proxy targets).*
>
> To empirically validate our method under realistic, imperfect proxy conditions and test its sensitivity to the Neural Collapse (NC) hypothesis, we conducted a two-part experiment (see **Tables 1, 2, and 3** in the attached PDF).
>
> **Part 1: Quantifying Baseline Proxy Alignment (Table 1)**
> * **Purpose:** To determine the actual geometric relationship between pre-trained features and classifier weights prior to unlearning.
> * **Metrics & Results:** We computed the Mean Alignment (average cosine similarity between sample embeddings and their target proxy weight) and Alignment Variance. While perfect collapse is not observed (means range from 0.11 to 0.29), the **Alignment Variance is exceptionally low** ($\approx 0.002$). This proves individual samples cluster tightly and uniformly around their shared proxy weights, providing a stable geometric target.
>
> **Part 2: Zero-Shot Graceful Degradation Analysis (Tables 2 & 3)**
> * **Purpose:** To test behavior when proxy alignment is inherently poor, we injected increasing Gaussian noise ($\sigma \in \{0.0, 0.1, 0.2, 0.3\}$) into the target weights to artificially degrade the proxies.
> * **Metrics & Results:** As $\sigma$ increases and targets become less reliable, retain utility is **significantly negatively affected** (e.g., CIFAR-10 retain accuracy drops from 93.2% to 71.5% in class unlearning). However, **unlearning efficacy is not affected**: Forget Accuracy remains near 0%, and RMIA safely drops, maintaining strong privacy.
>
> **Conclusion:** These results demonstrate that our method is effectively **sensitive to the Neural Collapse hypothesis being true**. If the proxy alignment is poor, the method still successfully erases the targeted concepts, but incurs a severe and proportional penalty in retain performance. We will incorporate these findings and the graceful degradation analysis into the revised manuscript.
>
> ---
> **W2 & Q2: Verifiability and Privacy Evaluations in Class Unlearning**
>
> We agree that low forget accuracy alone does not guarantee that information is fully unrecoverable. However, standard privacy evaluations like Membership Inference Attacks (MIA) are fundamentally ill-suited for the class unlearning scenario.
>
> Because a full class is removed in this setting, it becomes trivial for an attack model to distinguish between test samples (which belong to the retain classes) and forget samples. Since they belong to very different distributions (in-distribution retain classes vs. the out-of-distribution forget class), the model's prediction confidence naturally differs drastically between the two sets.
>
> Consequently, MIA scores in class unlearning are heavily skewed by this distribution shift and provide a misleading privacy signal. This is exactly why we rigorously evaluate and report our privacy metrics (such as MIA and RMIA) in the **Random Unlearning** setting. In that scenario, the underlying class distributions remain intact, making membership inference a valid, strict measure of whether specific data points were truly unlearned.

---

> > ### Author Rebuttal · Reviewer_RoUx · 2026-04-03
> >
> > Thank you for the detailed rebuttal and the additional experiments. I appreciate the added proxy alignment diagnostics and the noise based degradation analysis. I also appreciate the clear explanation of why standard MIA style evaluations may be unsuitable for the class unlearning setting.
> >
> > However, my concern is only partially resolved. The low mean alignment explicitly shows that there is a significant geometric gap between the actual pre-trained feature centers and the classifier weights. Therefore, the current evidence is insufficient to fully prove that classifier weights are reliable proxies. Does this inherent gap serve as a fundamental bottleneck for the retain utility in your zero-shot formulation? Is this utility penalty an unavoidable trade-off of the zero-shot setting? For this reason, I consider the concern partially resolved rather than fully resolved.
> >
> > I appreciate the authors' constructive and substantive response. If these points can be further clarified in the revision, I would be open to increasing my score. At the same time, the rebuttal does reinforce my current positive assessment, and I expect to maintain at least my current positive rating.

---

> > > ### Author Response · Authors · 2026-04-06
> > >
> > > We thank the reviewer for this thoughtful follow-up. We agree with the main point: the originally reported sample-level alignment statistics appeared low and, by themselves, do not fully establish classifier weights as reliable proxies for class-conditional structure. Concretely, the quantity we previously called "Mean Alignment" is the same metric we now denote as `sample_cos_mean`.
> > >
> > > Because these values were indeed modest, we revisited the question using the original Neural Collapse framework of Papyan, Han, and Donoho, *Prevalence of neural collapse during the terminal phase of deep learning training*, PNAS 2020 ([doi:10.1073/pnas.2015509117](https://doi.org/10.1073/pnas.2015509117)). In this framework, the most relevant quantities are not sample-level cosine statistics, but within-class collapse (NC1), alignment between classifier weights and class means (NC3), and agreement between the learned classifier and the nearest-class-center rule (NC4). These metrics more directly test whether the final-layer weights can serve as class prototypes.
> > >
> > > We therefore computed the following baseline Neural Collapse diagnostics:
> > >
> > > - `NC1`: within-class collapse relative to between-class separation; lower means each class is better summarized by a single prototype.
> > > - `NC3`: alignment between classifier weights and empirical class means; higher means the weights better match class prototypes.
> > > - `NC4`: agreement between the learned classifier and the nearest-class-center rule; higher means the classifier behaves more like a prototype-based classifier.
> > > - `acc_gap`: difference between classifier and nearest-class-center training accuracy; lower means less utility is lost when replacing the classifier by class prototypes.
> > >
> > > | Dataset | `NC1` ↓ | `NC3` ↑ | `NC4` ↑ | `acc_gap` ↓ |
> > > |---|---:|---:|---:|---:|
> > > | CIFAR-10 | 0.2116 | 0.8188 | 0.9996 | 0.0004 |
> > > | CIFAR-100 | 2.0760 | 0.7462 | 0.9908 | 0.0090 |
> > > | Tiny-ImageNet | 7.8577 | 0.7005 | 0.9355 | 0.0643 |
> > >
> > > These results clarify why `sample_cos_mean` alone is too indirect. It conflates two effects: misalignment between classifier weights and class means, and dispersion of samples around their class means. By contrast, `NC3` directly measures alignment between each classifier weight $w_y$ and empirical class mean $\mu_y$, `NC1` measures how well each class is summarized by a single prototype, and `NC4` together with `acc_gap` provides the strongest functional test of whether the classifier behaves as a prototype-based rule.
> > >
> > > To answer your two questions directly:
> > >
> > > **1. Does this inherent gap serve as a fundamental bottleneck for retain utility in the zero-shot formulation?**
> > >
> > > Yes. The bottleneck is real, but it is better characterized by the Neural Collapse metrics above than by `sample_cos_mean` alone. On CIFAR-10, the proxy is nearly exact: despite a modest sample cosine, `NC4=0.9996` and `acc_gap=0.0004`, indicating that the classifier weights behave almost identically to class prototypes in practice. On CIFAR-100, this agreement remains strong but is weaker (`NC4=0.9908`, `acc_gap=0.0090`). On Tiny-ImageNet, the proxy remains reasonably aligned but is less exact (`NC4=0.9355`, `acc_gap=0.0643`), together with worse `NC1`, showing that the representation is less collapsed and therefore less well captured by a single prototype. Thus, the geometric gap is a fundamental bottleneck, and its severity depends on how strongly the representation exhibits Neural Collapse.
> > >
> > > **2. Is this utility penalty an unavoidable trade-off of the zero-shot setting?**
> > >
> > > Yes, in the strict data-free setting. Without access to retain examples, one cannot compute the true empirical class centers and must instead rely on quantities already stored in the model, namely the classifier weights. Any residual mismatch between $w_y$ and the true class-conditional geometry therefore defines an intrinsic upper bound on zero-shot retain utility. The table suggests that this trade-off is mild on simpler datasets such as CIFAR-10 and becomes more noticeable as the class structure becomes less collapsed. So we agree with the reviewer’s concern in substance: the utility penalty is not merely an artifact of the metric choice, but a real consequence of the zero-shot regime, although the stronger NC diagnostics show that this penalty is smaller than sample cosine alone would suggest.
> > >
> > > In the revision, we will include these additional Neural Collapse diagnostics in an appendix. They clarify two points: (i) the degree to which the pre-trained representation fulfills Neural Collapse is an intrinsic bottleneck for strictly zero-shot unlearning, since the method must rely on classifier weights as proxies for empirical class centers; and (ii) in our experiments this fulfillment is overall medium-to-high, although it degrades consistently from CIFAR-10 to CIFAR-100 to Tiny-ImageNet. We will make this limitation explicit in the final paper.

---

### Decision · Program_Chairs · 2026-04-30

**Decision:**

Accept (regular)

**Comment:**

This paper proposes a black-box unlearning method targeting the output space. The proposed method is novel, and the paper has provided complete empirical evaluation, especially considering those added during the rebuttal phase. One concern for the paper is that the unlearning does not change the encoder but only operates at the output level. It may conflict with the principle of general unlearning settings in which privacy is important. The paper needs to clarify its scope in a revised version.